# Analysis of diverse eukaryotes suggests the existence of an ancestral mitochondrial apparatus derived from the bacterial type II secretion system

Lenka Horváthová [1,15], Vojtěch Žárský[1,15], Tomáš Pánek[2,14,15], Romain Derelle[3], Jan Pyrih[4,5], Alžběta Motyčková[1], Veronika Klápšťová[1], Martina Vinopalová [1], Lenka Marková[1], Luboš Voleman[1], Vladimír Klimeš[2], Markéta Petrů[1], Zuzana Vaitová[1], Ivan Čepička[6], Klára Hryzáková [7], Karel Harant[8], Michael W. Gray [9], Mohamed Chami[10], Ingrid Guilvout[11], Olivera Francetic [11], B. Franz Lang[12], Čestmír Vlček[13], Anastasios D. Tsaousis [4], Marek Eliáš [2✉] & Pavel Doležal [1✉]

The type 2 secretion system (T2SS) is present in some Gram-negative eubacteria and used to secrete proteins across the outer membrane. Here we report that certain representative heteroloboseans, jakobids, malawimonads and hemimastigotes unexpectedly possess homologues of core T2SS components. We show that at least some of them are present in mitochondria, and their behaviour in biochemical assays is consistent with the presence of a mitochondrial T2SS-derived system (miT2SS). We additionally identified 23 protein families co-occurring with miT2SS in eukaryotes. Seven of these proteins could be directly linked to the core miT2SS by functional data and/or sequence features, whereas others may represent different parts of a broader functional pathway, possibly also involving the peroxisome. Its distribution in eukaryotes and phylogenetic evidence together indicate that the miT2SS-centred pathway is an ancestral eukaryotic trait. Our findings thus have direct implications for the functional properties of the early mitochondrion.

[1] Faculty of Science, Department of Parasitology, Charles University, BIOCEV, Vestec, Czech Republic. [2] Faculty of Science, Department of Biology and Ecology, University of Ostrava, Ostrava, Czech Republic. [3] School of Biosciences, University of Birmingham, Edgbaston, UK. [4] Laboratory of Molecular & Evolutionary Parasitology, RAPID group, School of Biosciences, University of Kent, Canterbury, UK. [5] Institute of Parasitology, Biology Centre, Czech Academy of Sciences, České Budějovice, Czech Republic. [6] Faculty of Science, Department of Zoology, Charles University, Prague 2, Czech Republic. [7] Faculty of Science, Department of Genetics and Microbiology, Charles University, Prague 2, Czech Republic. [8] Faculty of Science, Proteomic core facility, Charles University, BIOCEV, Vestec, Czech Republic. [9] Department of Biochemistry and Molecular Biology and Centre for Comparative Genomics and Evolutionary Bioinformatics, Dalhousie University, Halifax, NS, Canada. [10] Center for Cellular Imaging and NanoAnalytics, University of Basel, Basel, Switzerland. [11] Biochemistry of Macromolecular Interactions Unit, Department of Structural Biology and Chemistry, Institut Pasteur, CNRS UMR3528, Paris, France. [12] Robert Cedergren Centre for Bioinformatics and Genomics, Département de Biochimie, Université de Montréal, Montreal, QC, Canada. [13] Institute of Molecular Genetics, Czech Academy of Sciences, Prague 4, Czech Republic. [14] Present address: Faculty of Science, Department of Zoology, Charles University, Prague 2, Czech Republic. [15] These authors contributed equally: Lenka Horváthová, Vojtěch Žárský, Tomáš Pánek. ✉email: marek.elias@osu.cz; pavel.dolezal@natur.cuni.cz

Mitochondria of all eukaryotes arose from the same Alphaproteobacteria-related endosymbiotic bacterium[1,2]. New functions have been incorporated into the bacterial blueprint during mitochondrial evolution, while many ancestral traits have been lost. Importantly, in some cases, these losses occurred independently in different lineages of eukaryotes, resulting in a patchy distribution of the respective ancestral mitochondrial traits in extant eukaryotes. Examples are the ancestral mitochondrial division apparatus (including homologues of bacterial Min proteins), the aerobic-type rubrerythrin system, or the tmRNA-SmpB complex, each retained in different subsets of distantly related protist lineages[3-5]. It is likely that additional pieces of the ancestral bacterial cell physiology will be discovered in mitochondria of poorly studied eukaryotes.

An apparent significant difference between the mitochondrion and bacteria (including those living as endosymbionts of eukaryotes) lies in the directionality of protein transport across their envelope. All bacteria export specific proteins from the cell via the plasma membrane using the Sec or Tat machineries[6], and many diderm (Gram-negative) bacteria exhibit specialised systems mediating further protein translocation across the outer membrane (OM)[7]. In contrast, the mitochondrion depends on a newly evolved protein import system spanning both envelope membranes and enabling import of proteins encoded by the nuclear genome[8]. The capacity of mitochondria to secrete proteins seems to be limited. Mitochondrial homologues of Tat translocase subunits occur in some eukaryotic taxa, but their role in protein secretion has not been established[9,10]. A mitochondrial homologue of the SecY protein (a Sec translocase subunit) has been described only in jakobids[11,12] and its function remains elusive[13]. No dedicated machinery for protein export from the mitochondrion across the outer mitochondrial membrane has been described.

One of the best characterised bacterial protein translocation machineries is the so-called type 2 secretion system (T2SS)[14,15]. The T2SS belongs to a large bacterial superfamily of type 4 pili (T4P)-related molecular machines, most of which secrete long extracellular filaments (pili) for motility, adhesion or DNA uptake[16-18]. The T2SS constitutes a specialised secretion apparatus, whose filament (pseudopilus) remains in the periplasm[14,15]. It is composed of 12–15 conserved components, commonly referred to as general secretion pathway (Gsp) proteins, which assemble into four main subcomplexes (Fig. 1A). The OM pore is formed by an oligomer of 15–16 molecules of the GspD protein[19,20]. The subcomplex in the inner membrane (IM) is called the assembly platform and consists of the central polytopic membrane protein GspF surrounded by single-pass membrane proteins GspC, GspL and GspM[21]. GspC links the assembly platform to the OM pore by interacting with the periplasmic N-terminal domain of GspD[22,23]. The third subcomplex, called the pseudopilus, is a helical filament formed mainly of GspG subunits, with minor pseudopilins (GspH, GspI, GspJ and GspK) assembled at its tip[24]. Pseudopilus assembly from its inner membrane base is believed to push the periplasmic T2SS substrate through the OM pore. The energy for pseudopilus assembly is provided by the fourth subcomplex, the hexameric ATPase GspE, interacting with the assembly platform from the cytoplasmic side[25,26]. Substrates for T2SS-mediated secretion are first transported by the Tat (as folded proteins) or the Sec (in an unfolded form) system across the IM into the periplasm, where they undergo maturation and/or folding. The folded substrates are finally loaded onto the pseudopilus for the release outside the cell via the OM pore. The known T2SS substrates differ among taxa and share no common sequence or structural features. Proteins transported by the T2SS in different species include catabolic enzymes (such as lipases, proteases or phosphatases) and, in the case of bacterial pathogens, toxins[14]. A recent survey of bacterial genomes showed that the T2SS is mainly present in Proteobacteria[18]. Crucially, neither the T2SS nor other systems of the T4P superfamily have been reported from eukaryotes[7,14,18].

Here we show that certain distantly related eukaryotes unexpectedly contain homologues of key T2SS subunits representing all four functional T2SS subcomplexes. We provide evidence for mitochondrial localisation of some of these eukaryotic Gsp protein homologues and describe experimental results supporting the idea that they constitute a system similar to the bacterial T2SS. Furthermore, we point to the existence of 23 proteins with a perfect taxonomic co-occurrence with the eukaryotic Gsp homologues. Some of these co-occurring proteins seem to be additional components of the mitochondrial T2SS-related machinery, whereas others are candidates for components of a broader functional pathway linking the mitochondrion with other parts of the cell. Given its phylogenetic distribution, we propose that the discovered pathway was ancestrally present in eukaryotes. Its further characterisation may provide fundamental insights into the evolutionary conversion of the proto-mitochondrion into the mitochondrial organelle.

## Results

**Certain protist lineages code for a conserved set of homologues of T2SS core components.** While searching the genome of the heterolobosean *Naegleria gruberi* for proteins of bacterial origin with a possible mitochondrial role, we surprisingly discovered homologues of four core subunits of the bacterial T2SS, specifically GspD, GspE, GspF, and GspG (Fig. 1a and Supplementary Data 1). Using genomic and transcriptomic data from public repositories and our on-going sequencing projects for several protist species of key evolutionary interest (see Methods section), we mapped the distribution of these four components in eukaryotes. All four genes were found in the following characteristic set of taxa (Fig. 1b and Supplementary Data 1): three additional heteroloboseans (*Naegleria fowleri*, *Neovahlkampfia damariscottae*, *Pharyngomonas kirbyi*), two jakobids (*R. americana* and *Andalucia godoyi*) and two malawimonads (*Malawimonas jakobiformis* and *Gefionella okellyi*). In addition, single-cell transcriptomes from two species of Hemimastigophora (hemimastigotes[27]) revealed the presence of homologues of GspD and GspG (*Hemimastix kukwesjijk*) and GspG only (*Spironema* cf. *mulitciliatum*), possibly reflecting incompleteness of the data. Finally, three separate representatives of the heterolobosean genus *Percolomonas* (Supplementary Fig. 1) each exhibited a homologue of GspD, but not of the remaining Gsp proteins, in the available transcriptomic data. In contrast, all four genes were missing in sequence data from all other eukaryotes investigated, including the genome and transcriptome of another malawimonad ("*Malawimonas californiana*") and deeply-sequenced transcriptomes of a third jakobid (*Stygiella incarcerata*) and four additional heteroloboseans (*Creneis carolina*, *Dactylomonas venusta*, *Harpagon schusteri*, and the undescribed strain Heterolobosea sp. BB2).

Probing *N. gruberi* nuclei with fluorescence in situ hybridization (FISH) ruled out an unidentified bacterial endosymbiont as the source of the *Gsp* genes (Supplementary Fig. 2). Moreover, the eukaryotic *Gsp* genes usually have introns and constitute robustly supported monophyletic groups well separated from bacterial homologues (Fig. 2 and Supplementary Fig. 3), ruling out bacterial contamination in all cases. In an attempt to illuminate the origin of the eukaryotic Gsp proteins we carried out systematic phylogenetic analyses based on progressively expanded datasets of prokaryotic homologues and for each tree inferred the taxonomic identity of the bacterial ancestor of the eukaryotic

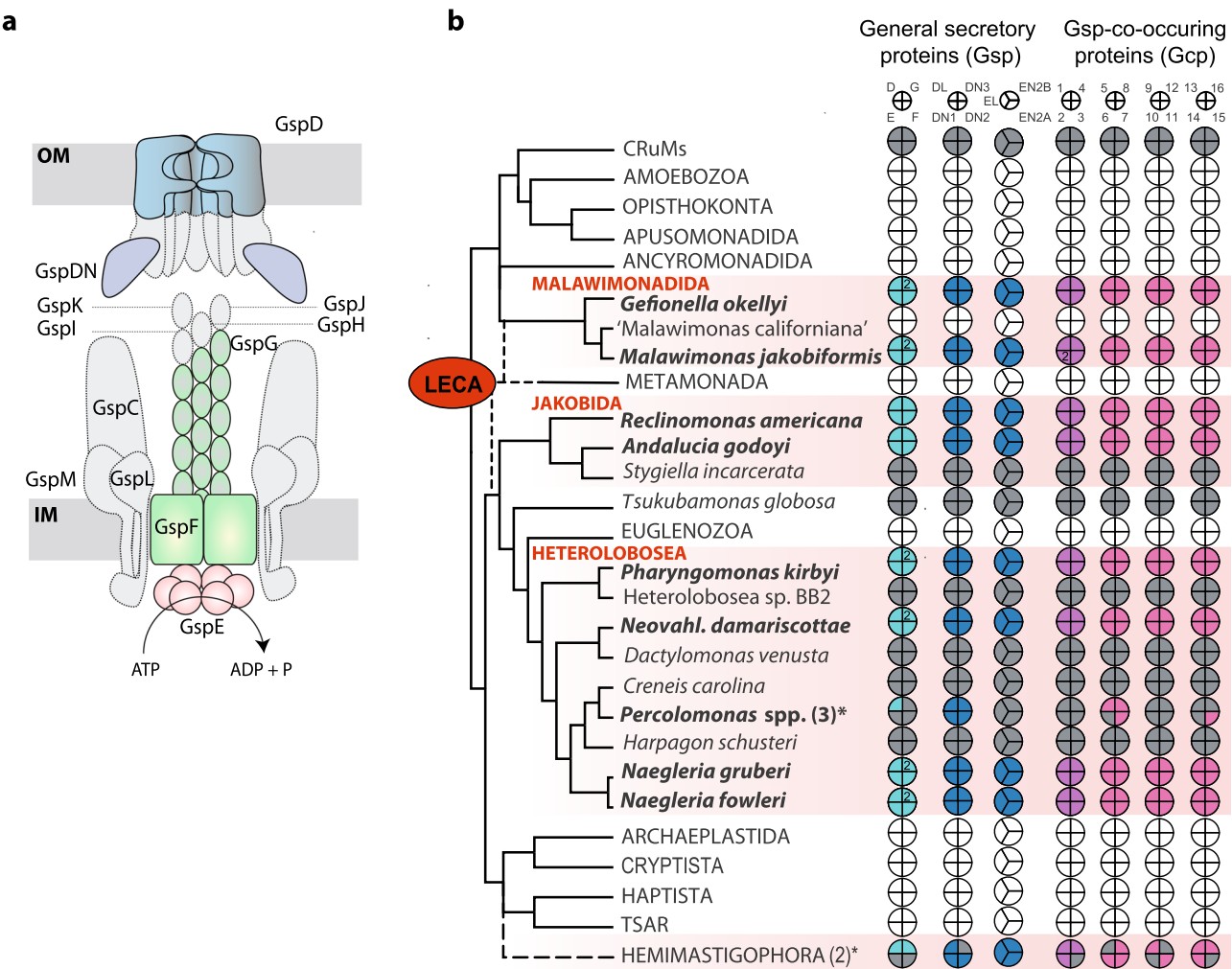

**Fig. 1 Some eukaryotes harbour homologues of core components of the bacterial T2SS machinery. a** Schematic representation of the complete bacterial T2SS; subunits having identified eukaryotic homologues are highlighted in colour. For simplicity, GspDN represents in the figure three different eukaryotic proteins (GspDN1 to GspDN3), together corresponding to two versions of a conserved domain present as a triplicate in the N-terminal region of the bacterial GspD protein. **b** Phylogenetic distribution of eukaryotic homologues of bacterial T2SS subunits (Gsp proteins) and co-occurring proteins (Gcp). Core T2SS components (cyan), eukaryote-specific T2SS components (dark blue), Gcp proteins carrying protein domains found in eukaryotes (magenta) and Gcp proteins without discernible homologues or with homologues only in prokaryotes (pink). Coloured sections indicate proteins found to be present in genome or transcriptome data; white sections, proteins absent from complete genome data; grey sections, proteins absent from transcriptome data. The asterisk indicates the presence of the particular protein in at least two of the three *Percolomonas* or at least one of the two Hemimastigophora species analysed. The species name in parentheses has not yet been formally published. Sequence IDs and additional details on the eukaryotic Gsp and Gcp proteins are provided in Supplementary Data 1. The tree topology and taxon names reflect most recent phylogenomic studies of eukaryotes;[32,101,102] the root (LECA) is placed according to Derelle et al.[29].

branch (see Methods section for details on the procedure). The results, summarised in Supplementary Fig. 3, showed that the inference is highly unstable depending on the dataset analysed, and no specific bacterial group can be identified as an obvious donor of the eukaryotic Gsp genes. This result probably stems from a combination of factors, including the long branches separating the eukaryotic and bacterial Gsp sequences, the length of Gsp proteins restricting the amount of the phylogenetic signal retained, and perhaps also rampant horizontal gene transfer (HGT) of the T2SS system genes between bacterial taxa. The eukaryotic Gsp genes are in fact so divergent that some of them could not be unambiguously classified as specific homologues of T2SS components (rather than the related machineries of the T4P superfamily) when analysed using models developed for the bacterial genomes[18] (Supplementary Fig. 3).

Heteroloboseans, jakobids and malawimonads have been classified in the supergroup Excavata[28]. However, recent

phylogenomic analyses indicate that excavates are non-monophyletic and even suggest that malawimonads are separated from heteroloboseans and jakobids by the root of the eukaryote phylogeny[29–32]. Together with the presence of at least some Gsp proteins in hemimastigotes, which constitute an independent eukaryotic supergroup[27], it is likely that the T2SS-related proteins were present in the last eukaryotic common ancestor (LECA) but lost in most eukaryote lineages (Fig. 1b). Heteroloboseans and malawimonads have two GspG paralogues, but the phylogenetic analyses did not resolve whether this is due to multiple independent GspG gene duplications or one ancestral eukaryotic duplication followed by loss of one of the paralogues in jakobids (Fig. 2; Supplementary Fig. 3D; and Supplementary Data 1).

**The eukaryotic Gsp proteins localise to the mitochondrion.** We hypothesised that the eukaryotic homologues of the four Gsp proteins are parts of a functional T2SS-related system localised to

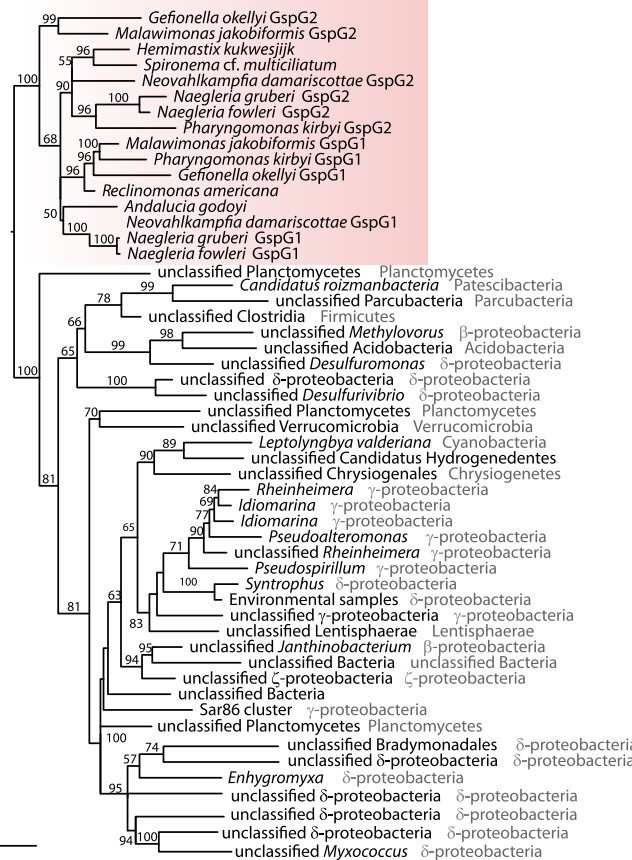

**Fig. 2 Eukaryotic Gsp homologues are monophyletic.** Maximum likelihood (ML) phylogenetic tree of eukaryotic and selected bacterial GspG proteins demonstrating the monophyletic origin of the eukaryotic GspG proteins and their separation from bacterial homologues by a long branch (the tree inferred using IQ-TREE). Branch support (bootstrap) was assessed by ML ultrafast bootstrapping and is shown only for branches where support is >50.

the mitochondrion. This notion was supported by the presence of predicted N-terminal mitochondrial targeting sequences (MTSs) in some of the eukaryotic Gsp proteins (Supplementary Data 1). The prediction algorithms identified putative N-terminal MTSs for proteins from jakobids and malawimonads but failed to recognise them in the orthologues from heteroloboseans, which, however, carry the longest N-terminal extensions (Supplementary Fig. 4).

In order to test if Gsp proteins carry functional mitochondrial targeting information, we expressed GspD, GspE and GspG1 genes from *N. gruberi* (*Ng*) and *G. okellyi* (*Go*) in *Saccharomyces cerevisiae*; no expression of *GspF* was achieved. All proteins were specifically localised in mitochondria, as confirmed by their co-localisation with Mitotracker red CMX Ros (Fig. 3a). Additionally, we attempted to express these genes in *Trypanosoma brucei*, which represents the evolutionarily closest experimental model to the eukaryotes carrying the *Gsp* genes. Of all the proteins tested, only *Go*GspD and *Go*GspG2 were detected (Fig. 3b), in addition to a weak signal for *Ng*GspG1 (Supplementary Fig. 5A). While both GspG proteins could be found specifically in the mitochondrion of *T. brucei*, *Go*GspD was found in a different membrane compartment, perhaps due to mistargeting. Finally, we tested if the atypically long N-terminal extension of *Ng*GspG1 targets the protein to the *T. brucei* mitochondrion. Indeed, the 160 N-terminal amino acid residues of *Ng*GspG1 were able to deliver the marker (mNeonGreen) into the organelle

(Supplementary Fig. 5B). As a complementary approach, we raised specific polyclonal antibodies against *Ng*GspG1 and probed *N. gruberi* cellular fractions (Fig. 3c and Supplementary Fig. 6). *Ng*GspG1 and *Ng*GspEN2A co-fractionated with the mitochondrial markers including alternative oxidase (AOX) and TatC[9], but not with the cytosolic protein hemerythrin[33]. Immunofluorescence microscopy of *N. gruberi* with the anti-*Ng*GspG1 antibody provided further evidence that *Ng*GspG1 is targeted to mitochondria (Fig. 3d and Supplementary Fig. 7).

In order to further confirm the mitochondrial localisation of the Gsp proteins in *N. gruberi*, we analysed the mitochondrial proteome of this organism by partial purification of the organelle and identification of resident proteins by mass spectrometry. A mitochondria-enriched fraction was obtained from a cellular lysate by several steps of differential centrifugation and OptiPrep gradient centrifugation. Three sub-fractions of different densities, named accordingly as Opt1015, Opt1520, Opt2030, were collected (Supplementary Fig. 8A, see Methods section for more details), with the densest one being most enriched in mitochondria. The sub-fractions were subjected to proteomic analysis. The relative amount of each protein in each sub-fraction was determined by label-free quantification and the proteins were grouped by a multicomponent analysis (for details see Methods section) according to their distributions across the gradient (Fig. 3e). A set of marker proteins (homologues of well-characterised typical mitochondrial proteins from other species) was used to identify a cluster of mitochondrial proteins. Due to the partial co-purification of peroxisomes with mitochondria, a peroxisome-specific cluster was defined analogously. As a result, 946 putative mitochondrial and 78 putative peroxisomal proteins were identified among the total of 4198 proteins detected in all three sub-fractions combined. Encouragingly, the putative mitochondrial proteome of *N. gruberi* is dominated by proteins expected to be mitochondrial or whose mitochondrial localisation is not unlikely (Supplementary Fig. 8B and Supplementary Data 2). On the other hand, the putative peroxisomal proteome seems to be contaminated by mitochondrial proteins (owing to the presence of several mitochondrial ribosomal proteins; Supplementary Data). Importantly, all five Gsp proteins (including both GspG paralogues) were identified in the putative mitochondrial but not peroxisomal proteome of *N. gruberi*.

**The properties of the eukaryotic Gsp proteins support the existence of a mitochondrial T2SS-related machinery.** The foregoing experiments support the idea that all four eukaryotic Gsp homologues localise to and function in the mitochondrion. However, direct in vivo demonstration of the existence of a functional mitochondrial T2SS-related machinery is currently not feasible, because none of the Gsp homologue-carrying eukaryotes represents a tractable genetic system. We thus used in vitro approaches and heterologous expression systems to test the key properties of the eukaryotic Gsp proteins.

Crucial for the T2SS function is the formation of the OM pore, which is a β-barrel formed by the oligomerization of the C-domain of the GspD protein[34]. The actual assembly of the bacterial pore requires GspD targeting to the outer membrane through interaction of its very C-terminal domain (S-domain) with the outer membrane lipoprotein GspS[35]. GspD forms a pre-pore multimer, whose OM membrane insertion is independent on the β-barrel assembly machinery (BAM) complex[36,37]. In addition, the bacterial GspD carries four short N-terminal domains exposed to the periplasm, called N0 to N3, of which N1 to N3 share a similar fold[38] (Fig. 4a). While the N3 domain is required for the pore assembly[39], N0 interacts with GspC of the assembly platform[22,40]. Sequence analysis of the mitochondrial

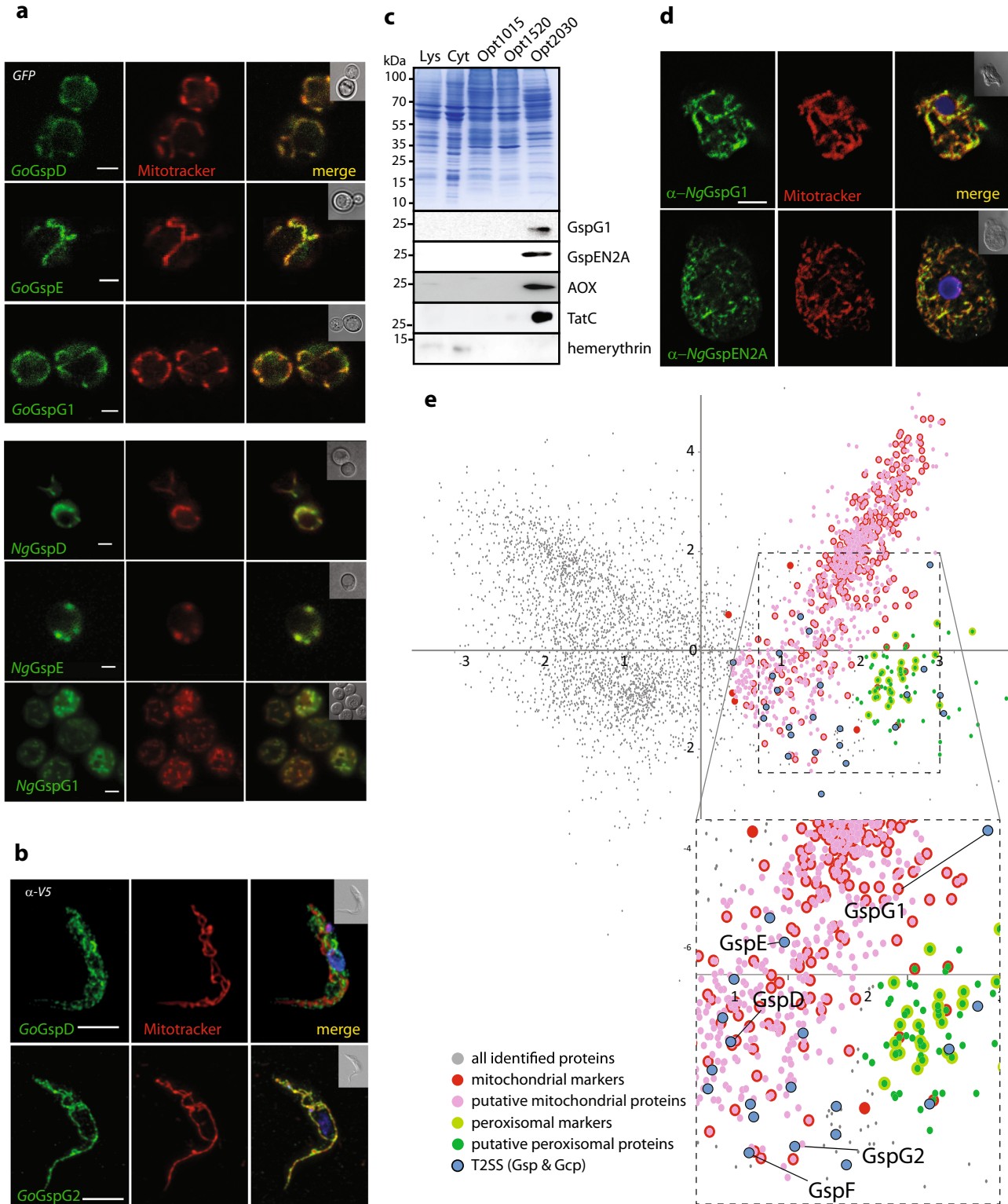

**Fig. 3 Eukaryotic T2SS components are localised in mitochondria. a** *S. cerevisiae* expressing *G. okellyi* and *N. gruberi* Gsp proteins as C-terminal GFP fusions stained with MitoTracker red CMX ROS to visualise mitochondria. **b** Expression of *G. okellyi* GspD and GspG2 with the C-terminal V5 Tag in *T. brucei* visualised by immunofluorescence; the cells are co-stained with MitoTracker red. **c** Cellular fractions of *N. gruberi* labelled by specific polyclonal antibodies raised against GspG1, GspEN2A and mitochondrial (alternative oxidase – AOX, TatC,) cytosolic (hemerythrin) marker proteins. **d** Immunofluorescence microscopy of *N. gruberi* labelled with specific polyclonal antibodies raised against GspG1 and GspEN2A, and co-stained with MitoTracker red. Scale bar (parts **a**–**d**), 10 μm. (for **a**–**d**, representative images of multiple, at least three, experiments are shown), **e** PCA analysis of 4198 proteins identified in the proteomic analysis of *N. gruberi* subcellular fractions differentially enriched in mitochondria. The cluster of mitochondrial proteins was defined on the basis of 376 mitochondrial markers. The boundaries of the cluster of co-purified peroxisomal proteins were defined by 26 peroxisomal markers.

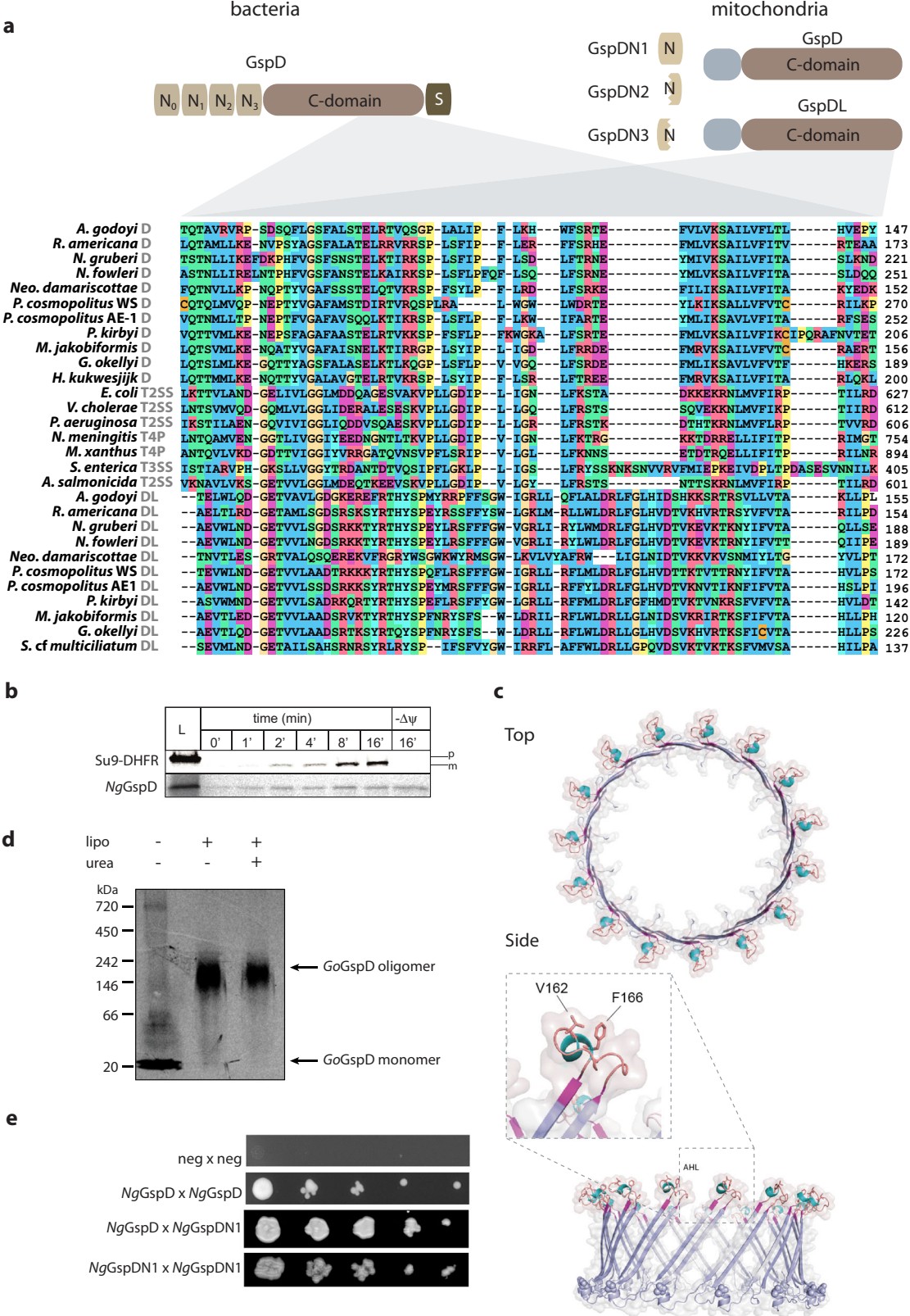

GspD homologue revealed that it corresponds to a C-terminal part of the bacterial GspD C-domain, whereas the N-terminal domains N0 to N3, the N-terminal part of the C-domain, and the S-domain are missing (Fig. 4a and Supplementary Fig. 4A).

According to our hypothesis, the mitochondrial GspD should be present in the outer mitochondrial membrane. In order to test

this localisation, *N. gruberi* GspD (*Ng*GspD) was in vitro imported into yeast mitochondria. The import reaction was also performed with the widely used synthetic substrate Su9-DHFR destined to mitochondrial matrix, which is composed of 69 amino acid residues of $F_o$-ATPase subunit 9 from *Neurospora crassa* fused to the mouse DHFR[41]. Both proteins accumulated in the

**Fig. 4 Mitochondrial GspD oligomerizes towards the formation of membrane pores. a** Domain architecture of the canonical bacterial GspD and the eukaryotic proteins homologous to its different parts (short N-terminal region of mitochondrial GspD of unidentified homology shown in grey). Below, protein sequence alignment of the secretin C-domain of bacterial and mitochondrial orthologues (mitochondrial GspD or GspDL and the respective molecular complex of bacterial secretins is depicted in grey, the numbers on the right depict the position of the amino acid in the particular sequence). **b** In vitro import of NgGspD into isolated yeast mitochondria over a period of 16 min. Dissipation of the membrane potential (ΔΨ) by AVO mix abolished the import of matrix reporter protein (Su9-DHFR) bud did not affect the mitochondrial GspD; p precursor of Su9-DHFR, m mature form of the protein upon cleavage of the mitochondrial targeting sequence. **c** Structural model of GoGspD built by ProMod3 on the Vibrio cholerae GspD template. Top and side view of a cartoon and a transparent surface representation of the GoGspD pentadecamer model is shown in blue. The amphipathic helical loop (AHL), a signature of the secretin family, is highlighted and coloured according to the secondary structure with strands in magenta, helices in cyan and loops in light brown. The C-terminal GpsD residues are highlighted as spheres. The detailed view of the AHL region shows the essential residues V162 and F166 pointing towards the membrane surface. **d** In vitro translation and assembly of mitochondrial GoGspD into a high-molecular-weight complex; lipo liposomes added, urea liposome fraction after 2 M urea treatment. **e** Y2H assay suggests the self- and mutual interaction of NgGspD and NgGspDN1. (for **b**–**d**, representative images of three experiments are shown).

mitochondria in a time-dependent manner, but only the import of Su9-DHFR could be inhibited by the addition of ionophores, which dissipate membrane potential (ΔΨ; Fig. 4b). This result showed that the import of GspD is independent of ΔΨ, which is a typical feature of mitochondrial outer membrane proteins. The key question was if the mitochondrial GspD homologue has retained the ability to assemble into an oligomeric pore-forming complex. This possibility was supported by homology modelling of GspD from *G. okellyi* (*Go*GspD) using *Vibrio cholerae* GspD[42] as a template, which indicated that the protein has the same predicted fold as the typical secretin C-domain. Remarkably, the two transmembrane β-strands of *Go*GspD are separated by the highly conserved amphipathic helical loop (AHL; Fig. 4c) essential for secretin membrane insertion, suggesting its capability to form a pentadecameric pore complex. Indeed, radioactively labelled *Go*GspD assembled into a high-molecular-weight complex of ~200 kDa in an in vitro bacterial translation system in the presence of lecithin liposomes (Fig. 4d). The complex was resistant to 2 M urea treatment, which would remove non-specific protein aggregates, and was still pelleted with the liposomes, suggesting that it was inserted into the liposomes. These results showed that the mitochondrial GspD, despite being significantly truncated when compared to its bacterial homologues, has retained the major characteristics of bacterial secretins[19], including the capacity to form oligomers and insert into membranes.

To test if the mitochondrial GspD forms bona fide pores in the membrane, we aimed to produce His-tagged *Ng*GspD in *E. coli* and purify the membrane complexes for the electrophysiology analysis. Inducing the production of this protein in the bacterial cytoplasm was highly toxic (Fig. 5a). A similar phenomenon has been reported for bacterial secretins, which form pores in the inner bacterial membrane[19]. While this hampered protein production and purification, directing the GspD export to the periplasm by fusing it to the N-terminal signal peptide of *E. coli* DsbA protein alleviated the toxicity upon autoinduction. The His-*Ng*GspD variant was affinity-purified on a Ni-column followed by size exclusion chromatography, during which two peaks of about 230 kDa and 125 kDa could be observed (Fig. 5b and Supplementary Fig. 15). The pore-forming activity of His-*Ng*GspD purified from both protein peaks was demonstrated by conductivity measurements in black lipid membranes composed of an *E. coli* polar lipid extract. The channel recordings illustrated a very high stability of the inserted membrane pores (Fig. 5c). The amplitude histograms (Fig. 5d) suggested that the mitochondrial GspD can form variable arrangements resulting in stable membrane pores of different sizes.

The secretion mechanism of the T2SS relies on the assembly of pseudopilus made up of GspG subunits[43]. Comparison of the mitochondrial GspG with bacterial homologues revealed

important similarities as well as differences. These proteins share a complete pseudopilin domain preceded by a transmembrane domain, but the mitochondrial proteins are substantially longer due to extensions at both N- and C-termini (Fig. 6a). The N-terminal extension likely serves as a MTS, but the origin and function of the C-terminal extension (amounting to ~100 amino acid residues) is unclear, as it is well conserved among the mitochondrial GspG but lacks discernible homologues even when analysed by highly sensitive homology-detection methods (HMM-HMM comparisons with HHpred[44] and protein modelling using the Phyre2 server[45]). Structural modelling of the pseudopilin domain into the recently obtained cryo-EM reconstruction of the PulG (=GspG) complex from *Klebsiella oxytoca*[46] revealed the presence of key structural features in the mitochondrial GspG from *G. okellyi* (*Go*GspG1), supporting possible formation of a pseudopilus (Fig. 6b).

To test this directly, we purified a recombinant pseudopilin domain of *Ng*GspG1 under native conditions, which showed a uniform size of 25 kDa corresponding to the monomer (Fig. 7a). Possible protein-protein interaction of the purified pseudopilin domain was tested by thermophoresis of an NT-647-labelled protein. The measurements revealed specific self-interaction and plotting of the change in thermophoresis yielded a $K_d$ of 216 (±15.1) nM (Fig. 7b). Additionally, the interaction properties of GspG were followed by the bacterial two-hybrid assay (BACTH). When produced in bacteria, the mitochondrial *Go*GspG1 showed specific oligomerisation, typical of bacterial major pseudopilins[47], supporting its in vivo propensity to form a pseudopilus (Fig. 7c). In addition, *Go*GspG1 showed positive interaction with *Go*GspF, consistent with the analogous interaction of bacterial GspG with the IM-embedded GspF[47]. Moreover, the mitochondrial *Go*GspF and *Go*GspE each formed dimers in the BACTH assay (Fig. 7c). These interactions are consistent with the hypothesised role of both proteins as mitochondrial T2SS components, as GspF forms dimers within the IM complex and GspE assembles into an active hexameric ATPase in the bacterial T2SS system. Indeed, bacterial GspG and GspF also interact in the BACTH assay[47]. Tests of all other possible interactions of *G. okellyi* Gsp proteins were negative.

The in silico analyses and experiments described above are consistent with the hypothesised existence of a functional mitochondrial secretion machinery derived from the bacterial T2SS. However, the mitochondrial subunits identified would assemble only a minimalist version of the secretion system, reduced to the functional core of the four subcomplexes of the bacterial T2SS, i.e., the luminal ATPase (GspE), the IM pseudopilus assembly platform (GspF), the intermembrane space pseudopilus (GspG) and the OM pore (truncated GspD). Despite using sensitive HMM-based searches, we did not detect homologues of other conserved T2SS subunits in any of the

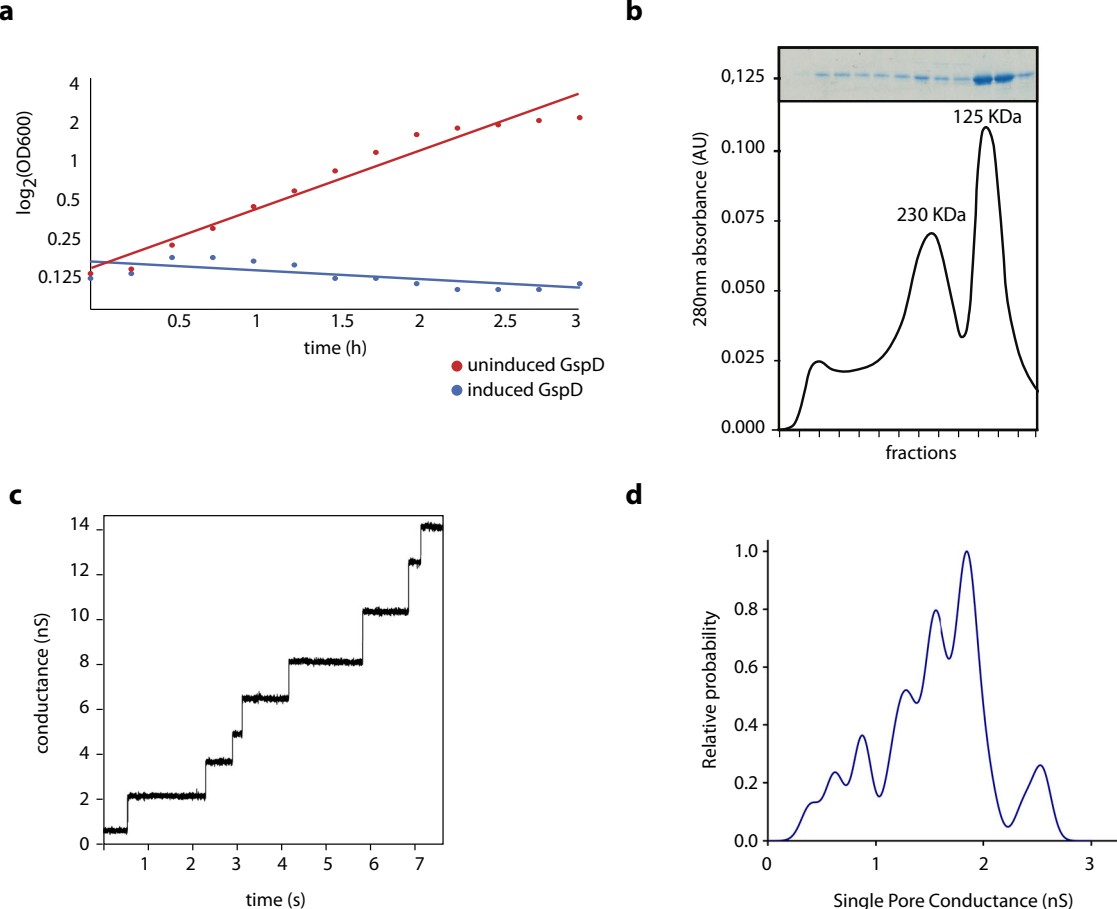

**Fig. 5 Membrane pore formation of mitochondrial GspD. a** Expression of mitochondrial NgGspD in bacteria quickly causes cell death upon induction. **b** Size exclusion chromatography of NgGspD shows two peaks of putative GspD complexes (bottom) and SDS–PAGE of the corresponding fractions (top). **c** Single channel recording of the NgGspD complex. The applied membrane potential was 15 mV. **d** Histogram of different recorded amplitudes during channel opening indicates multiple types of NgGspD-formed pores in the membrane (For **a–d**, representative images of three experiments are shown).

eukaryotes possessing GspD to GspG proteins. One of the missing subunits is GspC, which connects the assembly platform with the N0 domain of GspD pore[22,23]. Thus, the absence of GspC in eukaryotes correlates with the lack of the N0 domain in the eukaryotic GspD. Analogously, the absence of the C-terminal S-domain in the mitochondrial GspD (Fig. 4a), known to be missing also from some bacterial GspD proteins, rationalises the lack of a eukaryotic homologue of the bacterial OM component GspS that binds to GspD via the S-domain during the pore assembly[35]. The apparent lack of a eukaryotic homologue of GspL, which interacts via its cytosolic domain with the N1E domain of the bacterial GspE[48], may similarly be explained by the fact that the eukaryotic GspE protein seems to be homologous only to the C-terminal (CTE) domain of its bacterial counterpart and lacks an equivalent of the N1E domain (Supplementary Fig. 4e). The mitochondrial system also apparently lacks a homologue of GspO, a bifunctional enzyme that is essential for GspG maturation. Despite this absence, eukaryotic GspG homologues have conserved all the characteristic sequence features required for GspG maturation (the polar anchor and the transmembrane domain with a conserved glutamate residue at position +5 relative to the processing site; Fig. 6a and Supplementary Fig. 4J). Notably, all the NgGspG1 and NgGspG2-derived peptides detected in our proteomic analysis come from the region of the protein downstream of the conserved processing site (Fig. 6c), suggesting that analogous maturation of the pseudopilin also occurs in mitochondria.

**Additional putative components of the mitochondrial T2SS-based functional pathway identified by phylogenetic profiling.** Since none of the eukaryotes with the Gsp homologues is currently amenable to functional studies, we tried to further illuminate the role of the mitochondrial T2SS system using a comparative genomic approach. Specifically, we reasoned that possible additional components of the machinery, as well as its actual substrate(s), might show the same phylogenetic distribution as the originally identified four subunits. Using a combination of an automated identification of candidate protein families and subsequent manual scrutiny by exhaustive searches of available eukaryote sequence data (for details of the procedure see Methods section), we identified 23 proteins (more precisely, groups of orthologues) that proved to exhibit the same phylogenetic distribution in eukaryotes as the four core T2SS components. Specifically, all 23 proteins were represented in each of the heterolobosean, jakobid, and malawimonad species possessing all four core Gsp proteins, whereas only 17 proteins were identified in the incomplete transcriptomic data of hemimastigotes and seven of them were found in the transcriptomic data from the *Percolomonas* lineage that possesses only GspD (Fig. 1b and Supplementary Data 1). Except for two presumably Gsp-positive jakobids represented by incomplete EST surveys and a case of a likely contamination (Supplementary Data 3), no orthologues of any of these proteins were found in any other eukaryote (including the Gsp-lacking members of heteroloboseans, jakobids and malawimonads). The sequences of these 23 proteins were

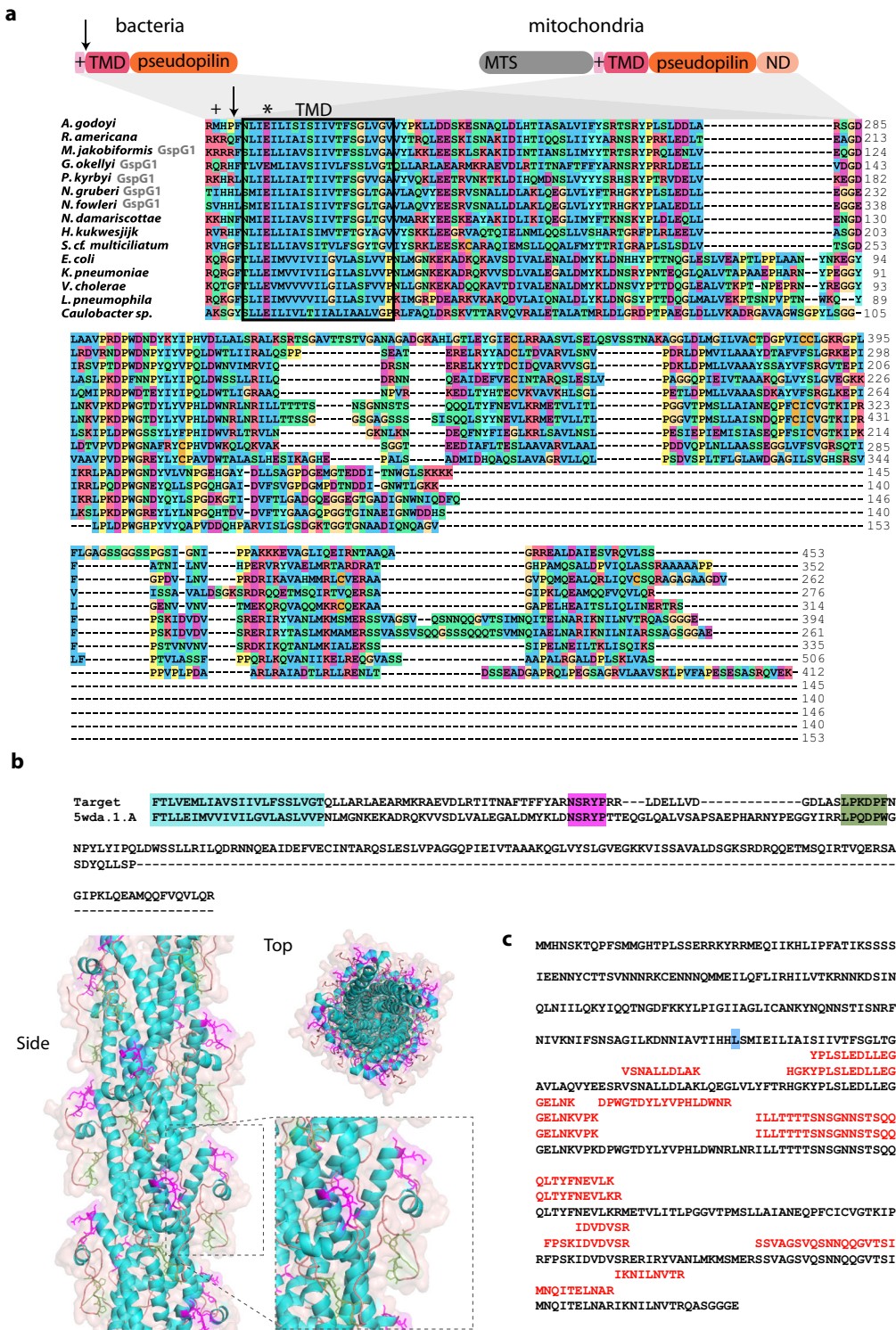

**Fig. 6 Structure and maturation of mitochondrial GspG. a** Domain architecture of the bacterial and the mitochondrial pseudopilin GspG (top). The arrow indicates the processing site of the bacterial GspG during protein maturation. MTS mitochondria targeting sequence, + polar anchor, TMD transmembrane domain, ND novel domain. (Bottom) Protein sequence alignment of the pseudopilin domains of mitochondrial and bacterial GspG proteins (in case two paralogues are present, only GspG1 is shown). **b** Homology modelling of *Go*GspG1. Top: pairwise alignment of protein sequences of *Go*GspG1 and *Klebisella oxytoca* PulG (the template used in model building). Regions of high sequence similarity are highlighted, including the hydrophobic segment (cyan), the α-β loop (magenta) and GspG signature loop with conserved Pro residues (green). Bottom: side and top views of the cartoon and transparent surface representation of *Go*GspG pilus model based on the pseudopilus cryo-EM reconstruction[46]. Proteins are coloured based on the secondary structure, with helix regions in cyan and loops in light brown. The regions of GspG1 sharing high similarity with PulG are highlighted with the same colour code as in the sequence alignment, with side chains shown in magenta and green. Inset: detail of the structurally conserved loop regions. The novel domain (ND) specific to mitochondrial GspG proteins was omitted from the modelling. **c** Peptides specific to *Ng*GspG1 retrieved from the proteomic analysis of *N. gruberi* mitochondria. The Leu residue highlighted in blue indicates residue +1 following the processing site of bacterial GspG proteins.

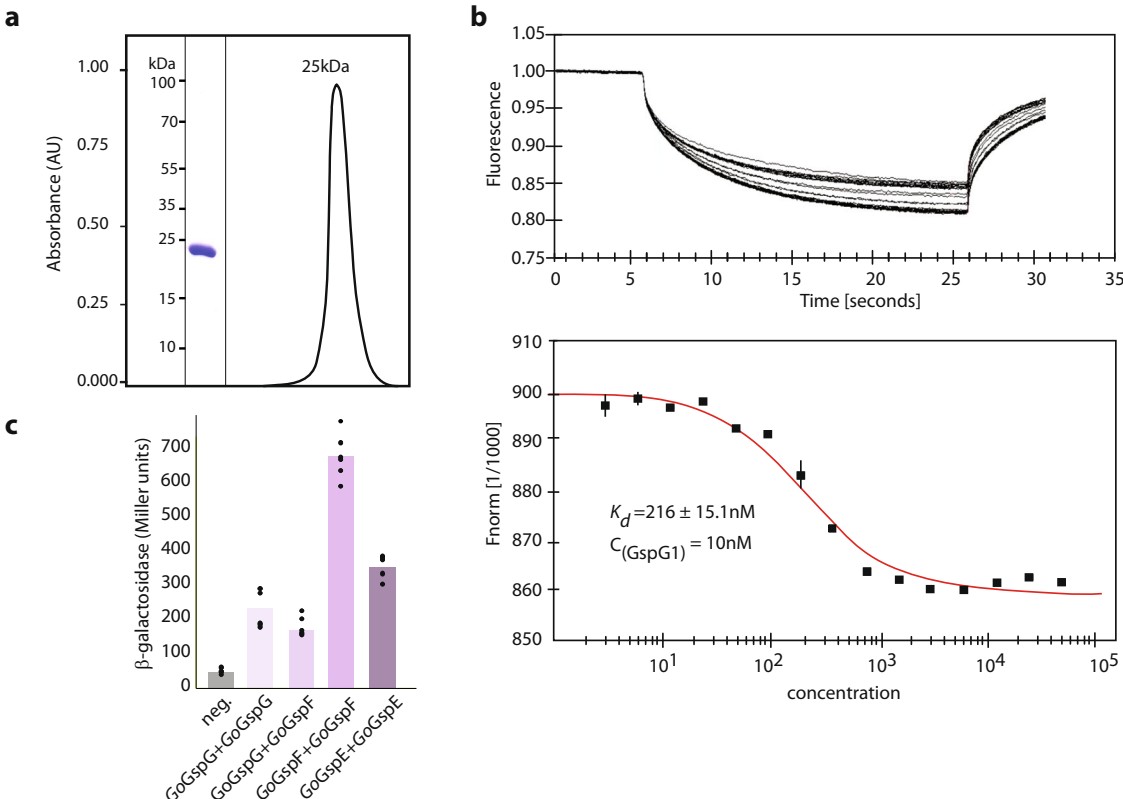

**Fig. 7 The protein interactions of mitochondrial GspG. a** The soluble part of the pseudopilin domain was purified to homogeneity on and ran through a gel filtration column, which demonstrated the stable monomeric state of the protein (representative image of three experiments is shown). **b** Thermophoresis of NT-647-labelled protein showed specific self-interaction of the domain (top); plotting of the change in thermophoresis yielded a $K_d$ of 216 ± 15.1 nm, concentration of the labelled GspG was 10 nM (bottom), The error bars depict standard deviation; $n = 3$. **c** Positive interaction between the mitochondrial GspG protein and other mitochondrial T2SS subunits as determined by the BACTH assay, $n = 6$.

analysed by various in silico approaches, including HHpred and Phyre2 to assess their possible function (Fig. 8a).

These analyses revealed that seven of the families have a direct link to the T2SS suggested by discerned homology to known T2SS components. One of them represents an additional, more divergent homologue of the C-terminal part of the bacterial GspD. Hence, the protein has been marked as GspDL (GspD-like). Three other families, referred to as GspDN1 to GspDN3, proved to be homologous to the Secretin_N domain (Pfam family PF03958), present in the bacterial GspD protein as domains N1, N2 and N3 (Fig. 4a). The N1-N3 array protrudes into the periplasm, where it oligomerizes to form three stacked rings[20]. As mentioned above, the initially identified eukaryotic GspD homologues lack the N-terminal region, suggesting that the gene was split into multiple parts in eukaryotes. While GspDN1 corresponds to a full single N-domain, GspDN2 and GspDN3 relate only to its C-terminal and N-terminal halves, respectively (Fig. 4a and Supplementary Fig. 4B–D). Unfortunately, high sequence divergence makes it impossible to identify potential specific correspondence between the N1 to N3 domains of the bacterial GspD and the eukaryotic GspDN1 to GspDN3 proteins. Importantly, a Y2H assay indicated that the two separate polypeptides GspD and GspDN1 of *N. gruberi* may interact in vivo (Fig. 4e), perhaps forming a larger mitochondrial complex. In addition, we identified most of the discovered GspD-related proteins (GspDL/N) in the *N. gruberi* mitochondrial proteome (the exception being GspDN1, which was not detected in a sufficient number of replicates to be included in the downstream analysis; Supplementary Data 2).

The final three proteins linked to the T2SS based on their sequence features seem to be evolutionarily derived from GspE. One, denoted GspEL (GspE-like) represents a divergent homologue of the C1E (i.e. nucleotide-binding) domain of GspE, although with some of the characteristic motifs (Walker A, Asp box, Walker B) abrogated (Supplementary Fig. 4F), indicating the loss of the ATPase function. The other two proteins, which we denote GspEN2A and GspEN2B, are suggested by HHpred to be related to just the N2E domain of the bacterial GspE (Supplementary Fig. 4G, H). GspEN2A and GspEN2B were identified among *N. gruberi* mitochondrial proteins in the proteomic analysis, whereas GspEL was found in the cluster of putative peroxisomal proteins. Importantly, a polyclonal antibody raised against NgGspEN2A confirmed the mitochondrial localisation of the protein (Fig. 3c, d and Supplementary Figs. 6 and 7).

The remaining sixteen proteins co-occurring with the core eukaryotic T2SS subunits, hereafter referred to as Gcp (<u>G</u>sp-<u>c</u>o-occurring <u>p</u>roteins), were divided into three categories. The first comprises four proteins that constitute paralogues within broader common eukaryotic (super)families. Three of them (Gcp1 to Gcp3) belong to the WD40 superfamily and seem to be most closely related to the peroxisomal protein import co-receptor Pex7 (Supplementary Fig. 9). None of these proteins has any putative N-terminal targeting sequence, but interestingly, the peroxisomal targeting signal 1 (PTS1) could be predicted on most Gcp1 and some Gcp2 proteins (Supplementary Data 1). However, these predictions are not fully consistent with the results of our proteomic analysis: NgGcp1 was found among the mitochondrial proteins and NgGcp2 in the cluster of putative peroxisomal proteins (Supplementary Data 2), but PTS1 is predicted to be

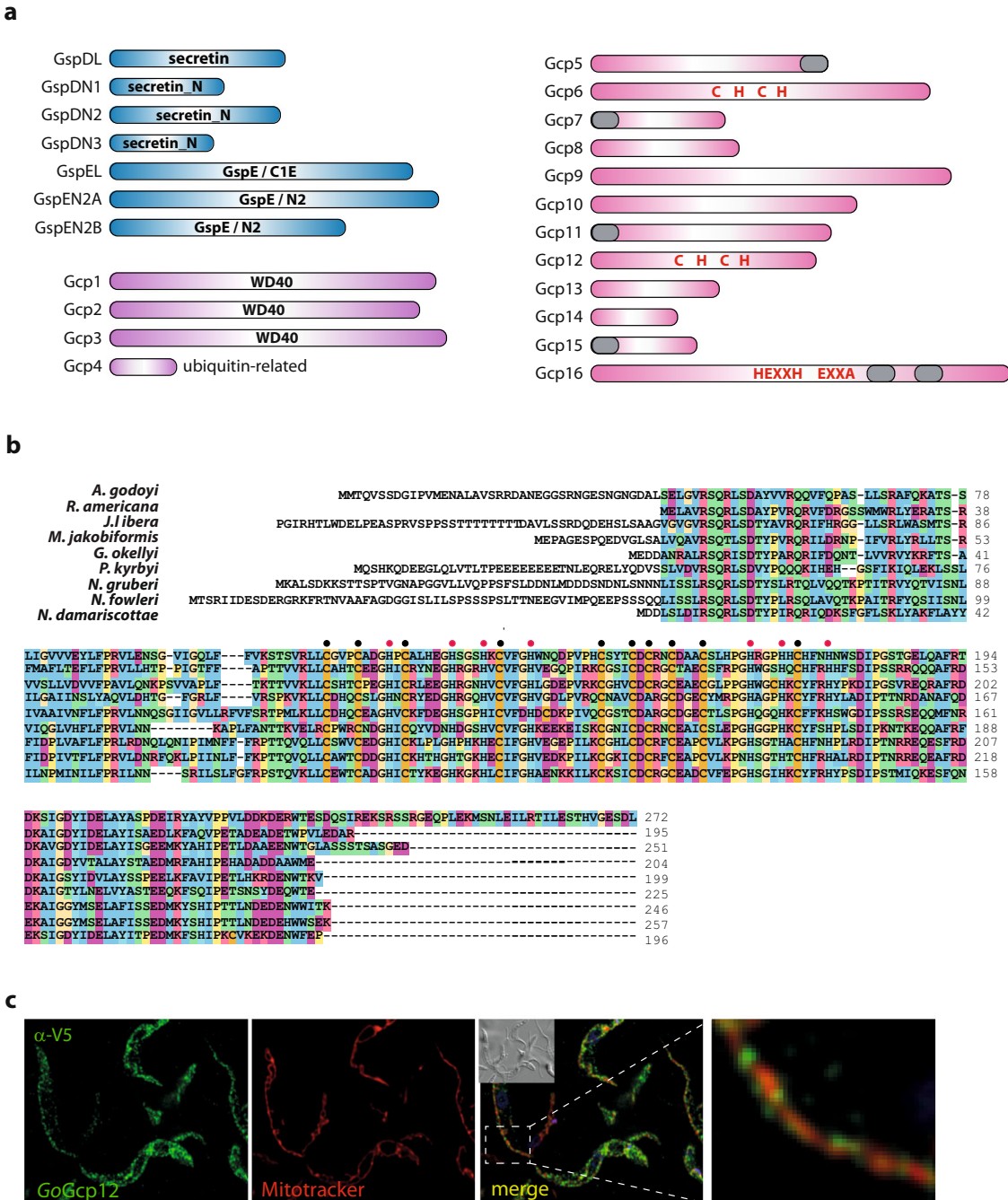

**Fig. 8 Proteins with the same phylogenetic profile as the originally identified mitochondrial Gsp homologues. a** Schematic domain representation of 23 proteins occurring in heteroloboseans, jakobids and malawimonads with the core T2SS subunits but not in other eukaryotes analyzed. Proteins with a functional link to the T2SS suggested by sequence homology are shown in blue, proteins representing novel paralogues within broader (super) families are shown in violet, and proteins without discernible homologues or with homologues only in prokaryotes are shown in pink. The presence of conserved protein domains or characteristic structural motifs is shown if detected in the given protein. Grey block – predicted transmembrane domain (see also Supplementary Fig. 11); "C H C H" – the presence of absolutely conserved cysteine and histidine residues (see also Supplementary Fig. 12) that may mediate binding of a prosthetic group; "HEXXH" and "EXXA" in Gcp16 indicate absolutely conserved motifs suggesting that the protein is a metallopeptidase of the gluzincin group (see text). The length of the rectangles corresponds to the relative size of the proteins. **b** Protein sequence alignment of Gcp12 proteins with highlighted conserved cysteine (black circles) and histidine (red circles) residues. **c** The expression of *Go*Gcp12 in *T. brucei* with the C-terminal V5 tag (green) showed partial co-localisation with the mitochondrion (red) (representative image of three experiments is shown). Scale bar 10 μm.

present in the *Ng*Gcp1 protein (Supplementary Data 1). The fourth Gcp protein (Gcp4) is a paralogue of the ubiquitin-like superfamily, distinctly different from the previously characterised members including ubiquitin, SUMO, NEDD8 and others (Supplementary Fig. 10).

The second Gcp category comprises eleven proteins (Gcp5 to Gcp15) well conserved at the sequence level among the Gsp-containing eukaryotes, yet lacking any discernible homologues in other eukaryotes or in prokaryotes. Two of these proteins (Gcp8, Gcp15) were not identified in the proteomic analysis of *N. gruberi*

(Supplementary Data 1 and 2). Of those identified, several (Gcp5, Gcp6, Gcp13) were found among the mitochondrial proteins, whereas some others (Gcp9, Gcp10, Gcp11) clustered with peroxisomal markers. Specific localisation of the three remaining proteins (Gcp7, Gcp12, and Gcp14) could not be determined due to their presence at the boundaries of the mitochondrial or peroxisomal clusters. No homology to other proteins or domains could be discerned for the Gsp5 to Gsp15 proteins even when sensitive homology-detection algorithms were employed. However, four of them are predicted as single-pass membrane proteins, with the transmembrane segment in the N- (Gcp7, Gcp11, Gcp15) or C-terminal (Gcp5) regions (Fig. 8a and Supplementary Fig. 11). Interestingly, Gcp6 and Gcp12 proteins contain multiple absolutely conserved cysteine or histidine residues (Fig. 8a, b and Supplementary Fig. 12). We were not able to determine their localisation in *N. gruberi* by microscopy, but we tested the localisation of GoGcp12 upon expression in *T. brucei*, where it co-localised with the mitochondrial tubules (Fig. 8c).

Gcp16 constitutes a category of its own, as it typifies a newly described protein family present also in bacteria of the PVC superphylum (Supplementary Fig. 13). Phylogenetic analysis confirmed that the eukaryotic members of the family are of the same origin (Supplementary Fig. 14). Gcp16 proteins are predicted to harbour two transmembrane domains (Supplementary Fig. 13). Furthermore, HHpred searches suggested possible homology of a region of the Gcp16 protein (upstream of the transmembrane domains) to various metallopeptidases, although with inconclusive statistical support. However, inspection of the HHpred alignments revealed that Gcp16 shares with these hits an absolutely conserved motif HEXXH (Supplementary Fig. 13), which is the catalytic, metal-binding motif of the zincin tribe of metallopeptidases[49]. Interestingly, close to the HEXXH motif, Gcp16 possesses an absolutely conserved EXXA motif, which is diagnostic of a zincin subgroup called gluzincins[49], further supporting the notion that Gcp16 may function as a membrane-embedded peptidase. Most eukaryotic Gcp16 proteins exhibit an N-terminal extension compared to the bacterial homologues (Supplementary Fig. 13), but only some of these extensions are recognised as putative MTSs and the *N. gruberi* Gcp16 was not identified either in putative mitochondrial or peroxisomal proteome.

## Discussion

Our analyses revealed that a subset of species belonging to four eukaryotic lineages share a set of at least 27 proteins (or families of orthologues) absent from other eukaryotes for which genomic or transcriptomic data are currently available (Fig. 1B). At least eleven of these proteins (the Gsp proteins) are evolutionarily related to components of the bacterial T2SS, although seven of them are so divergent that their evolutionary connection to the T2SS could be recognised only retrospectively after their identification based on their characteristic phylogenetic profile. For the sixteen remaining proteins (Gcp1 to Gcp16) no other obvious evolutionary or functional link to the T2SS is evident apart from the same phyletic pattern as exhibited by the T2SS subunit homologues. Nevertheless, similar phylogenetic profiles are generally a strong indication for proteins being parts of the same functional system or pathway, and have enabled identification of additional components of different cellular structures or pathways (e.g. refs. [50,51]). Is it, therefore, possible that the 27 Gsp/Gcp proteins similarly belong to a single functional pathway?

The phylogenetic profile shared by the eukaryotic Gsp and Gcp proteins is not trivial, as it implies independent gene losses in a specific set of multiple eukaryotic branches (Fig. 1b). The

likelihood of a chance emergence of the same taxonomic distribution of these proteins is thus low. Nevertheless, false positives cannot be completely excluded among the Gcp proteins and their list may be revised when a more comprehensive sampling of eukaryote genomes or transcriptomes becomes available. It is also possible that the currently inferred phylogenetic profile of some of the Gsp/Gcp proteins is incomplete due to limited sampling of the actual gene repertoire of species represented by transcriptome assemblies only. The inherently incomplete nature of single-cell transcriptome assemblies available for hemimastigotes potentially explains our failure to identify homologues of some Gsp and Gcp proteins in this group (Fig. 1b and Supplementary Data 1). An incomplete set is evident also in the heterolobosean *Percolomonas* lineage, as transcriptomic data from three different members revealed only the presence of GspD, GspDL, the three GspDN proteins, and four Gcp proteins (Fig. 1b and Supplementary Data 1). The relatively coherent pattern of Gsp/Gcp protein occurrence in the three independently sequenced transcriptomes and the fact that in other Gsp/Gcp-containing eukaryotes (except for hemimastigotes) all 27 families are always represented in the respective transcriptome assembly (Supplementary Data 1) suggest that the *Percolomonas* lineage has indeed preserved only a subset of Gsp/Gcp families. Genome sequencing is required to test this possibility.

All uncertainties notwithstanding, our data favour the idea that a hitherto unrecognised complex functional pathway exists in some eukaryotic cells, underpinned by most, if not all, of the 27 Gsp/Gcp proteins and possibly others yet to be discovered. Direct biochemical and cell biological investigations are required for testing its existence and the actual cellular role. Nevertheless, we have integrated the experimental data gathered so far with the insights from bioinformatic analyses to propose a hypothetical working model (Fig. 9).

Our main proposition is that the eukaryotic homologues of the bacterial Gsp proteins assemble a functional transport system, here denoted miT2SS, that spans the mitochondrial OM and mediates the export of specific substrate proteins from the mitochondrion. Although the actual architecture of the miT2SS needs to be determined, the available data suggest that it departs in detail from the canonical bacterial T2SS organisation, as homologues of some of the important bacterial T2SS components are apparently missing. Most notable is the absence of GspC, presumably related to the modified structure of its interacting partner GspD, which in eukaryotes is split into multiple polypeptides and seems to completely lack the N0 domain involved in GspC binding. It thus remains unclear whether and how the IM assembly platform and the OM pore interact in mitochondria. One possible explanation is that GspC has been replaced by an unrelated protein. It is notable that three Gcp proteins (Gcp7, Gcp11 and Gcp15) have the same general architecture as GspC: they possess a transmembrane segment at the N-terminus and a (predicted) globular domain at the C-terminus (Fig. 9a and Supplementary Fig. 11). Testing possible interactions between these proteins and T2SS core subunits (particularly GspG, GspF and GspDN) using BACTH or Y2H assays will be of future interest.

Further investigations also must address the question of whether the mitochondrial GspG is processed analogously to the bacterial homologues and how such processing occurs in the absence of discernible homologues of GspO. The mitochondrial GspG is presumably inserted into the IM by the Tim22 or Tim23 complex, resulting in a GspG precursor with the N-terminus, including the MTS, protruding into the matrix. It is possible that N-terminal cleavage by matrix processing peptidase serves not only to remove the transit peptide, but at the same time to generate the mature N-terminus of the processed GspG form,

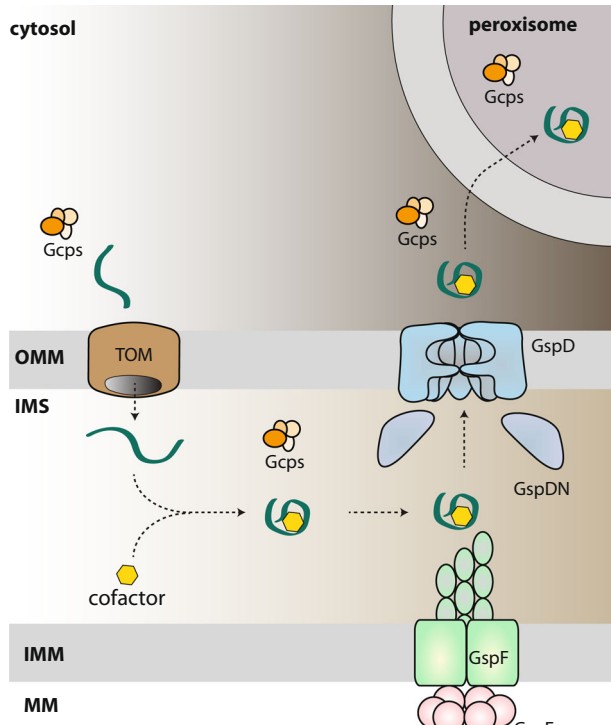

**Fig. 9 The mitochondrial T2SS (miT2SS) as part of a hypothetical eukaryotic functional pathway connecting the mitochondrion and the peroxisome.** The scheme presents the most reasonable interpretation of the findings reported in this study, but further work is needed to test details of the working model. According to the model, a nucleus-encoded protein (green), possibly one of the newly identified Gcp proteins, is imported via the TOM complex into the mitochondrial inner membrane space, where it is modified by addition of a specific prosthetic group. After folding it becomes a substrate of the miT2SS machinery, is exported from the mitochondrion and finally reaches the peroxisome. The loading of the prosthetic group, the delivery to the peroxisome and possibly also the actual function of the protein in the peroxisome is assisted by specific subsets of other Gcp proteins. The hypothetical presence of Gcp proteins in specific (sub)compartments is depicted as a group of orange ovals. OMM outer mitochondrial membrane, IMS intermembrane space, IMM inner mitochondrial membrane, MM mitochondrial matrix.

ready for recruitment into the pseudopilus. A different hypothesis is offered by the discovery of the Gcp16 protein with sequence features suggesting that it is a membrane-embedded metallo-peptidase (certainly non-homologous to GspO, which is an aspartic acid peptidase[52]). Although the subcellular localisation of Gcp16 needs to be established, we speculate that it might be a mitochondrial IM protein serving as an alternative prepilin peptidase.

In parallel with its apparent simplification, the miT2SS may have been specifically elaborated compared to the ancestral bacterial machinery. This possibility is suggested by the existence of two pairs of proteins corresponding to different parts of the bacterial GspE subunit. We propose that the eukaryotic GspE makes a heterodimer with the GspEN2A to reconstitute a unit equivalent to most of the bacterial GspE protein (lacking the GspL-interacting N1E domain), whereas GspEN2B, which is much more divergent from the standard N2E domain than GspEN2A, may pair specifically with GspEL to make an enzymatically inactive GspE-like version. We can only speculate as to the function of these proteins, but the fact that the bacterial GspE assembles into a homohexamer raises the possibility that in

eukaryotes catalytically active and inactive versions of GspE are mixed together in a manner analogous to the presence of catalytically active and inactive paralogous subunits in some well-known protein complexes, such as the proton-pumping ATPase (e.g. refs. [53,54]). The co-occurrence of two different paralogues of the GspD C-domain, one (GspDL) being particularly divergent, suggests a eukaryote-specific elaboration of the putative pore in the mitochondrial OM. Moreover, the electrophysiology measurements of the pores built of mitochondrial GspD indicated stable complexes of variable sizes; a property not observed for bacterial proteins. Finally, the C-terminal extension of the mito-chondrial GspG representing a conserved domain without discernible homology to other proteins suggests a eukaryote-specific modification of the pseudopilus functioning.

An unanswered key question is what is the actual substrate (or substrates) possibly exported from the mitochondrion by the miT2SS. No bioinformatic tool for T2SS substrate prediction is available due to the enigmatic nature of the mechanism of substrate recognition by the pathway[14], so at the moment we can only speculate. It is notable that no protein encoded by the mitochondrial genomes of jakobids, heteroloboseans and malawimonads stands out as an obvious candidate for the miT2SS substrate, since they either have well-established roles in the mitochondrion or are hypothetical proteins with a restricted (genus-specific) distribution. Therefore, we hypothesise that the substrate could be encoded by the nuclear genome and imported into the mitochondrion to undergo a specific processing/maturation step. This may include addition of a prosthetic group – a scenario modelled on the process of cytochrome *c* or Rieske protein maturation[55,56]. Interestingly, the proteins Gcp6 and Gcp12, each exhibiting an array of absolutely conserved cysteine and histidine residues (Supplementary Fig. 12), are good candidates for proteins to which a specific prosthetic group might be attached, so any of them could be the sought-after miT2SS substrate. Some of the other Gcp proteins may then represent components of the hypothetical machinery responsible for the substrate modification. The putative functionalization step may occur either in the mitochondrial matrix or in the intermembrane space (IMS), but we note that the former localisation would necessitate a mechanism of protein translocation across the mitochondrial IM in the matrix-to-IMS direction, which has not been demonstrated yet. Regardless, the T2SS system would eventually translocate the modified protein across the mito-chondrial OM to the cytoplasm.

However, this may not be the end of the journey, since there are hints of a link between the miT2SS-associated pathway and peroxisomes. First, three Gcp proteins, namely Gcp1 to Gcp3, are specifically related to Pex7, a protein mediating import of peroxisomal proteins characterised by the peroxisomal targeting signal 2 (PTS2)[57]. Second, some of the Gcp proteins (especially Gcp1 and Gcp13) have at the C-terminus a predicted PTS1 signal (at least in some species; Supplementary Data 1). Third, several Gcp proteins (Gcp2, Gcp9, Gcp10 and Gcp11) and GspEL were assigned to the putative peroxisomal proteome in our proteomic analysis (Supplementary Data 2). We note the discrepancy between the PTS1 signal predictions and the actual set of experimentally defined peroxisomal proteins, which might be due to an incomplete separation of peroxisome and mitochondria by our purification procedure, but may also reflect protein shuttling between the two organelles. We thus hypothesise that upon its export from the mitochondrion, the miT2SS substrate might be eventually delivered to the peroxisome. This is possibly mediated by the Gcp1/2/3 trio, but other Gcp proteins might participate as well. One such protein might be the ubiquitin-related protein Gcp4. Ubiquitination and deubiquitination of several components of the peroxisome protein import machinery are a critical

part of the import mechanism[57] and Gcp4 might serve as an analogous peptide modifier in the hypothetical peroxisome import pathway functionally linked to the miT2SS.

Altogether, our data suggest the existence of an elaborate functional pathway combining components of bacterial origin with newly evolved eukaryote-specific proteins. The extant phylogenetic distribution of the pathway is sparse, but our current understanding of eukaryote phylogeny suggests that it was ancestrally present in eukaryotes and for some reason dispensed with multiple times during evolution. Although we could not define a specific bacterial group as the actual source of the eukaryotic Gsp genes, it is tempting to speculate that the T2SS was introduced into eukaryotes by the bacterial progenitor of mitochondria and that it was involved in delivering specific proteins from the endosymbiont into the host cell, as is known in the case of current intracellular bacteria[58]. Elucidating the actual role of this communication route in establishing the endosymbiont as a fully integrated organelle requires understanding the cellular function of the modern miT2SS-associated pathways, which is a challenge for future research.

## Methods

**Sequence data and homology searches.** Homologues of relevant genes/proteins were searched in sequence databases accessible via the National Center for Biotechnology Information BLAST server (https://blast.ncbi.nlm.nih.gov/Blast.cgi), including the nucleotide and protein non-redundant (nr) databases, whole-genome shotgun assemblies (WGAs), expressed sequence tags (ESTs) and transcriptome shotgun assemblies (TSAs). Additional public databases searched included the data provided by the Marine Microbial Eukaryote Transcriptome Sequencing Project (MMETSP[59]) comprising TSAs from hundreds of diverse protists (https://www.imicrobe.us/#/projects/104), the OneKP project[60] (https://sites.google.com/a/ualberta.ca/onekp/) comprising TSAs from hundreds of plants and algae, and individual WGAs and TSAs deposited at various on-line repositories (Supplementary Data 4). To further improve the sampling, draft genome assemblies were generated in this study for the heterolobosean *Neovahlkampfia damariscottae* and the malawimonad informally called "*Malawimonas californiana*". Details on the sequencing and assembly are provided in Supplementary Methods. Finally, the analyses also included sequence data from genome and/or transcriptome projects for several protists that are underway in our laboratories and will be published in full elsewhere upon completion (Supplementary Data 4). Relevant sequences were extracted from these unpublished datasets and either deposited in GenBank or included in Supplementary Data 1.

Similarity searches were done using BLAST[61] (blastp or tblastn, depending on the database queried) and HMMER[62] using profile HMMs built from sequence alignments of proteins of interest. Hits were evaluated by BLAST (blastp or blastx) searches against the nr protein dataset at NCBI to distinguish orthologues of Gsp and Gcp proteins from paralogous proteins or non-specific matches. This was facilitated by a high degree of conservation of individual eukaryotic Gsp/Gcp proteins among different species (see also Supplementary Figs. 4 and 11–13) and in most cases by the lack of other close homologues in eukaryotic genomes (the exceptions being members of broader protein families, including the ATPase GspE, the WD40 superfamily proteins Gcp1 to Gcp3 and the ubiquitin-related protein Gcp4). All identified eukaryotic Gsp and Gcp sequences were carefully manually curated to ensure maximal accuracy and completeness of the data, which included correction of existing gene models, extension of truncated sequences by manual analysis of raw sequencing reads and correction of assembly errors (for details see Supplementary Methods). All newly predicted or curated Gsp and Gcp sequences are provided in Supplementary Data 1; additional Gsp and Gcp sequences from non-target species are listed in Supplementary Data 4. The nomenclature of the Gsp and Gcp genes proposed in this study was also reflected in the annotation of the *A. godoyi* genome, recently published as part of a separate study[5].

**Phylogenetic profiling.** In order to identify genes with the same phylogenetic distribution as the eukaryotic homologues of the four core T2SS components, we carried out two partially overlapping analyses based on defining groups of putative orthologous genes in select Gsp-positive species and phylogenetically diverse Gsp-negative eukaryotic species. The list of taxa included is provided in Supplementary Data 5. The first analysis was based on 18 species, including three Gsp-positive ones (*N. gruberi*, *A. godoyi* and *M. jakobiformis*), for the second analysis the set was expanded by adding one additional Gsp-positive species (*G. okellyi*) and one Gsp-negative species (*Monocercomonoides exilis*). Briefly, the protein sequences of a given species were compared to those of all other species using blastp followed by fast phylogenetic analyses, and orthologous relationships between proteins were then inferred from this set of phylogenetic trees using a reference-species-tree-independent approach. This procedure was repeated for each species and all resulting sets of orthologous relationships, also known as phylomes[63], were

combined in a dense network of orthologous relationships. This network was finally trimmed in several successive steps to remove weak or spurious connections and to account for (genuine or artificial) gene fusions, with the first analysis being less restrictive than the second. Details of this pipeline are provided in Supplementary Methods. For each of the two analyses, the final set of defined groups of orthologs (orthogroups) was parsed to identify those comprising genes from at least two Gsp-positive species yet lacking genes from any Gsp-negative species. The orthogroups passing this criterion were further analysed manually by blastp and tblastn searches against various public and private sequence repositories (see the section "Sequence data and homology searches") to exclude those orthogroups with obvious orthologs in Gsp-negative species. *Percolomonas* lineage exhibiting only GspD and jakobids represented by incomplete EST surveys (these species likely possess the miT2SS system) were not considered Gsp-negative. The orthogroups that remained were then evaluated for their conservation in Gsp-positive species and those that proved to have a representative in all these species (*N. gruberi*, *N. fowleri*, *N. damariscottae*, *P. kirbyi*, *A. godoyi*, *R. americana*, *M. jakobiformis*, *G. okellyi*) were considered as bona fide Gcp (Gsp-co-occurring protein) candidates. It is of note that some of these proteins are short and were missed by the automated annotation of some of the genomes, so using relaxed criteria for the initial consideration of candidate orthogroups (i.e. allowing for their absence from some of the Gsp-positive species) proved critical for decreasing the number of false-negative identifications.

**Sequence analyses and phylogenetic inference.** Subcellular targeting of Gsp and Gcp proteins was evaluated using TargetP-1.1 (ref. [64]; http://www.cbs.dtu.dk/services/TargetP-1.1/index.php), TargetP-2.0 (ref. [65]; http://www.cbs.dtu.dk/services/TargetP/), Mitoprot II (ref. [66]; https://ihg.gsf.de/ihg/mitoprot.html), MitoFates[67] (http://mitf.cbrc.jp/MitoFates/cgi-bin/top.cgi)[67], WoLF PSORT (https://wolfpsort.hgc.jp/) and PTS1 predictor[68] (http://mendel.imp.ac.at/pts1/). Transmembrane domains were predicted using TMHMM[69] (http://www.cbs.dtu.dk/services/TMHMM/). Homology of Gsp and Gcp protein families to other proteins was evaluated by searches against Pfam v. 31 (ref. [70]; http://pfam.xfam.org/) and Superfamily 1.75 database[71] (http://supfam.org/SUPERFAMILY/index.html), and by using HHpred[44] (https://toolkit.tuebingen.mpg.de/#/tools/hhpred) and the Phyre2 server[45] (http://www.sbg.bio.ic.ac.uk/phyre2/html/page.cgi?id=index). The relative position of the Gcp4 family among ubiquitin-like proteins was analysed by a cluster analysis using CLANS[72] (https://www.eb.tuebingen.mpg.de/protein-evolution/software/clans/); for the analysis the Gcp4 family was combined with all 59 defined families included in the clan Ubiquitin (CL0072) as defined in the Pfam database (each family was represented by sequences from the respective seed alignments stored in the Pfam database). For further details on the procedure see the legend of Supplementary Fig. 10A. Multiple sequence alignments used for presentation of the conservation and specific sequence features of Gsp and Gcp families were built using MUSCLE[73] and shaded using BioEdit (http://www.mbio.ncsu.edu/BioEdit/bioedit.html).

In order to obtain datasets for the phylogenetic analyses of eukaryotic GspD to GspG proteins, the protein sequences were aligned using MAFFT[74] and trimmed manually. Profile hidden Markov models (HMMs) built on the basis of the respective alignments were used as queries to search the UniProt database using HMMER. All recovered sequences were assigned to components of the T4P superfamily machineries using HMMER searches against a collection of profile HMMs reported by Abby et al. (ref. [75]). For each GspD to GspG proteins, a series of alignments was built by progressively expanding the sequence set by including more distant homologues (as retrieved by the HMMER searches). Specifically, the different sets of sequences were defined by the HMMER score based on the formula $score_{cutoff} = c * score_{best\ prokaryotic\ hit}$, with the coefficient c decreasing from 0.99 to 0.70 incrementally by 0.01. The sequences were then aligned using MAFFT, trimmed with BMGE[76] and the phylogenies were computed with IQ-TREE[77] using the best-fit model (selected by the programme from standard protein evolution models and the mixture models[78] offered). The topologies were tested using 10,000 ultra-fast bootstraps. The resulting trees were systematically analyzed for support of the monophyly of eukaryotic sequences and for the taxonomic assignment of the parental prokaryotic node of the eukaryotic subtree. The assignment was done using the following procedure. The tree was artificially rooted between the eukaryotic and prokaryotic sequences. From sub-leaf nodes to the deepest node of the prokaryotic subtree, the taxonomic affiliation of each node was assigned by proportionally considering the known or inferred taxonomic affiliations (at the phylum or class level) of the descending nodes. See the legend to Supplementary Fig. 3 for further details.

The phylogenetic analysis of the WD40 superfamily including Gcp1 to Gcp3 proteins was performed as follows. The starting dataset was prepared by a combination of two different approaches: (1) each identified sequence of Gcp1 to Gcp3 proteins was used as a query in a blastp search against the non-redundant (nr) NCBI protein database and the 500 best hits for each sequence were kept; (2) protein sequences of each of the Gcp1 to Gcp3 family were aligned using MAFFT and the multiple alignment was used as a query in a HMMER3 search (https://toolkit.tuebingen.mpg.de/#/tools/hmmer) against the UniProt database. Best hits (E-value cutoff 1e-50) from the two searches were pooled and de-duplicated, and the resulting sequence set (including Gcp1 to Gcp3 sequences) was aligned using MAFFT and trimmed manually to remove poorly conserved regions. Because WD40 proteins are very diversified, sequences that were too divergent were eliminated from the starting dataset during three subsequent

rounds of sequence removal, based on a manual inspection of the alignment and phylogenetic trees computed by IQ-TREE (using the best-fit model as described above). The final dataset was enriched by adding PEX7 and WDR24 orthologues from eukaryotes known to possess miT2SS components. The final phylogenetic tree was computed using IQ-TEE as described in the legend to Supplementary Fig. 9. IQ-TREE was used also for inferring trees of the heterolobosean 18 S rRNA gene sequences (Supplementary Fig. 1), ubiquitin-related proteins (Supplementary Fig. 10B) and the Gcp16 family (Supplementary Fig. 14); details on the analyses are provided in legends to the respective figures.

**Structural homology modelling.** The PDB database was searched by the SWISS-MODEL server[79] for structural homologues of GoGspD and GoGspG1. V. cholerae GspD[20] (PDB entry 5WQ9) and K. oxytoca PulG[46] pseudopilus (PDB entry 5WDA) were selected as the top matches, respectively. Models were built based on the target-template alignment using ProMod3 (Bienert et al.[79]). Coordinates that were conserved between the target and the template were copied from the template to the model. Insertions and deletions were remodelled using a fragment library, followed by rebuilding side chains. Finally, the geometry of the resulting model was regularised by using a force field. In the case of loop modelling with ProMod3 fails, an alternative model was built with PROMOD-II (Guex et al.[80]). The quaternary structure annotation of the template was used to model the target sequence in its oligomeric form[81].

**Cultivation and fractionation of N. gruberi and proteomic analysis.** Naegleria gruberi str. NEG-M was axenically cultured in M7 medium with PenStrep (100 U/mL of penicillin and 100 μg/mL of streptomycin) at 27 °C in vented tissue culture flasks. Mitochondria of N. gruberi were isolated in seven independent experiments and were analyzed individually (see below). Each time ~1 × 10^9 N. gruberi cells were resuspended in 2 mL of SM buffer (250 mM sucrose, 20 mM MOPS, pH 7.4) supplemented with DNase I (40 μg/mL) and Roche cOmplete™ EDTA-free Protease Inhibitor Cocktail and homogenised by eight passages through a 33-gauge hypodermic needle (Sigma Aldrich). The resulting cell homogenate was then cleaned of cellular debris by differential centrifugation and separated by a 2-hr centrifugation in a discontinuous density OptiPrep gradient (10%, 15%, 20%, 30 and 50%) as described previously[82]. Three visually identifiable fractions corresponding to 10–15% (OPT-1015), 15–20% (OPT-1520) and 20–30% (OPT-2023) OptiPrep densities were collected (each in five biological replicates) and washed with SM buffer.

Proteins extracted from these samples were then digested with trypsin and peptides were separated by nanoflow liquid chromatography and analyzed by tandem mass spectrometry (nLC-MS2) on a Thermo Orbitrap Fusion (q-OT-IT) instrument as described elsewhere[83]. The quantification of mass spectrometry data in the MaxQuant software[84] provided normalised intensity values for 4,198 proteins in all samples and all three fractions. These values were further processed using the Perseus software[85]. Data were filtered and only proteins with at least two valid values in one fraction were kept. Imputation of missing values, which represent low-abundance measurements, was performed with random distribution around the value of instrument sensitivity using default settings of Perseus software[85].

The data were analyzed by principle component analysis (PCA). The first two loadings of the PCA were used to plot a two-dimensional graph. Based on a set of marker proteins (376 mitochondrial and 26 peroxisomal, Supplementary Data 2), clusters of proteins co-fractionating with mitochondria and peroxisomes were defined and the proteins within the clusters were further analyzed. This workflow was set up on the basis of the LOPIT protocol[86]. As a result, out of the 4198 proteins detected, 946 putative mitochondrial and 78 putative peroxisomal proteins were defined. All proteins were subjected to in silico predictions concerning their function (BLAST, HHpred[44]) and subcellular localisation (Psort II, https://psort.hgc.jp/form2.html; TargetP, http://www.cbs.dtu.dk/services/TargetP/; MultiLoc2, https://abi.inf.uni-tuebingen.de/Services/MultiLoc2).

**Fluorescence in situ hybridization.** The PCR products of the NgGspE and NgGspF genes were labelled by alkali-stable digoxigenin-11-dUTP (Roche) using DecaLabel DNA Labeling Kit (Thermo Scientific). Labelled probes were purified on columns of QIAquick Gel Extraction Kit (Qiagen, 28704) in a final volume of 50 μL. Labelling efficiencies were tested by dot blotting with anti-digoxigenin alkaline phosphatase conjugate and CSPD chemiluminescence substrate for alkaline phosphatase from DIG High Prime DNA Labelling and Detection Starter Kit II (Roche) according to the manufacturer's protocol. FISH with digoxigenin-labelled probes was performed essentially according to the procedure described in Zuba-cova et al.[87] with some modifications. N. gruberi cells were pelleted by centrifugation for 10 min at 2000×g at 4 °C. Cells were placed in hypotonic solution, fixed twice with a freshly prepared mixture of methanol and acetic acid (3:1) and dropped on superfrost microscope slides (ThermoScientific). Preparations for hybridisations were treated with RNase A, 20 μg in 100 μL 2× SSC, for 1 h at 37 °C, washed twice in 2× SSC for 5 min, dehydrated in a methanol series and air-dried. Slides were treated with 50% acetic acid followed by pepsin treatment and post-fixation with 2% paraformaldehyde. Endogenous peroxidase activity of the cell remnants (undesirable for tyramide signal amplification) was inactivated by

incubation in 1% hydrogen peroxide, followed by dehydration in a graded methanol series. All slides were denatured together with 2 μL (25 ng) of the probe in 50 μL of hybridisation mixture containing 50% deionised formamide (Sigma) in 2× SSC for 5 min at 82 °C. Hybridisations were carried out overnight. Slides were incubated with tyramide reagent for 7 min. Preparations were counterstained with DAPI in VectaShield and observed under an Olympus IX81 microscope equipped with a Hamamatsu Orca-AG digital camera using the Cell^R imaging software.

**Heterologous gene expression, preparation of antibodies.** The selected Gsp genes from G. okellyi and N. gruberi were amplified from commercially synthesised templates (Genscript; for primers used for PCR amplification of the coding sequences see Supplementary Data 6) and cloned into the pUG35 vector. The constructs were introduced into S. cerevisiae strain YPH499 by the lithium acetate/PEG method. The positive colonies grown on SD-URA plates were incubated with MitoTracker Red CMXRos (Thermo Fisher Scientific) and observed for GFP and MitoTracker fluorescence (using the same equipment as used for FISH, see above). For the expression in T. brucei, sequences encoding full-length GoGspD, GoGspG2, NgGspG1 as well as the first 160 amino acid residues from NgGspG1 were amplified from the commercially synthesised templates and cloned into the pT7 plasmid, encoding either three C-terminal V5 tags (full-length genes) or C-terminal mNeonGreen followed by three V5 tags (NgGspG1 targeting sequence). T. brucei cell line SMOX 927 (Poon et al.[88]) were grown in SDM79 media[89] supplemented with 10% fetal bovine serum (Gibco). NotI-linearised plasmids (50 μg) were nucleofected into procyclic T. brucei cells using an Amaxa nucleofector (Lonza) as described before[90]. Expression of the genes was induced by an overnight incubation with doxycycline (1 μg/ml). For bacterial expression, genes encoding NgGspG1 and NgGspEN2A were amplified from commercially synthesised templates and cloned into the pET42b vector (for primers used for PCR amplification of the coding sequences, see Supplementary Data 6). The constructs were introduced into the chemically-competent E. coli strain BL21(DE3) and their expression induced by 1 mM IPTG. The recombinant proteins were purified under denaturing conditions on Ni-NTA agarose (Qiagen). The purified proteins were used for rat immunisation in an in-house animal facility at the Charles University.

**Immunofluorescence microscopy.** Yeast, procyclic T. brucei or N. gruberi cells were pre-treated by incubation with MitoTracker CMX Ros (1:000 dilution) for 20 min to stain mitochondria, washed twice in PBS and placed on coverslips. After a 5-min incubation, the cells were fixed with 4% PFA in PBS for 15 min. The solution was replaced by 0.1% Triton X-100 in PBS and the slides were incubated for 15 min. The slides were then treated with blocking buffer (1% BSA and 0.033% Triton X-100 in PBS) for 1 h at room temperature. After blocking, the samples were stained overnight at 4 °C with a blocking solution supplemented with the primary antibody (in-house-produced rat anti-NgGspG1 and anti-NgGspEN2A antibodies, dilutions 1:100, rat anti-V5 antibody Abcam, dilution 1:1000 dilution). The slides were washed three times for 10 min with 0.033% Triton X-100 and incubated with an anti-rat antibody conjugated with Alexa Fluor® 488 (1:1000 dilution, Thermo Fisher Scientific) in blocking buffer for 1 h at room temperature. Slides were washed twice in PBS supplemented with 0.033% Triton X-100 for 10 min followed by a single 10-min wash with PBS only. The slides were mounted in Vectashield containing DAPI (Vector laboratories). Static images were acquired on a Leica SP8 FLIM inverted confocal microscope equipped with 405 nm and white light (470–670 nm) lasers and a FOV SP8 scanner using an HC PL APO CS2 63x/1.4 NA oil-immersion objective. Laser wavelengths and intensities were controlled by a combination of AOTF and AOBS separately for each channel. Emitting fluorescence was captured by internal spectrally tunable HyD detectors. Imaging was controlled by the Leica LAS-X software. Images were deconvolved using the SVI Huygens Professional software (Scientific Volume Imaging) with the CMLE algorithm. Maximum intensity projections and brightness/contrast corrections were performed in the FIJI ImageJ software.

**Purification of native NgGspD and NgGspG.** His-tagged NgGspD carrying the signal peptide of E. coli DsbA was produced in the E. coli strain BL21 (DE3) in autoinduction media (50 mM Na₂HPO₄, 50 mM KH₂PO₄, 2% Tryptone, 0.5% Yeast extract, 85 mM NaCl, 0.5% glycerol, 0.05% glucose and 0.2% lactose) as described in Studier[91]. The cells were grown at 37 °C for 16 h, centrifuged at 6000×g for 15 min at 4 °C and resuspended in 20 mM Tris pH 8, 5 mM EDTA. Bacteria were incubated for 30 min on ice in the presence of lysozyme (1 mg/ml) and DNAse, and lysed in a French press. The cell lysate was centrifuged at 6000×g for 15 min at 4 °C to pellet cell debris. The cleared lysate was then centrifuged at 100,000×g for 2 h at 4 °C and membrane pellet was washed twice in 20 mM Tris pH 8, with 1-h centrifugation steps (100,000×g at 4 °C). The membranes were resuspended to the final protein concentration of 1 mg/ml in 50 mM Tris pH 8, 250 mM NaCl, 1% Zwittergent 3–14, and solubilized for 2 hr at 4 °C. The sample was then centrifuged at 100,000×g for 1 h at 4 °C. His-tagged proteins from the supernatant were incubated overnight with Ni-NTA agarose (Qiagen) at 4 °C. The next day, the agarose was collected on a column and washed with 50 ml of 50 mM Tris pH 8, 250 mM NaCl, 20 mM Imidazole, 0.5% Zwittergent 3–14. Bound proteins were eluted by 5 × 0.5 ml of 50 mM Tris pH 8, 250 mM NaCl, 250 mM imidazole, Zwittergent 3–14. Collected fractions were analyzed by SDS–PAGE and western

blotting. Selected samples were then pooled together, rebuffered into 50 mM Tris pH 8, 250 mM NaCl, 0.5% Zwittergent 3–14 and analyzed by size exclusion chromatography.

For NgGspG1, the BL21 cells expressing the pseudopilin domain lacking the N-terminal hydrophobic part were collected after 4 h of IPTG induction at 37 °C. The cells were collected, washed with PBS, and then resuspended in 35 ml of 50 mM Tris, 100 mM NaCl, pH 8 with added inhibitors (cOMPLETE™ tablets, EDTA-free, Roche), DNAse (1 mg/ml), 5 mM MgCl₂ and lysozyme (1 mg/ml). The suspension was incubated on ice for 30 min. After lysis via French press the resulting suspension was spun down for 20 min at $100,000 \times g$, 4 °C. Ni-NTA agarose beads (1 ml; Qiagen), washed and resuspended in loading buffer (50 mM Tris, 100 mM NaCl, 10 mM imidazole, pH 8), were added to the supernatant. The supernatant was incubated with the beads for 1 h on a tube rotator at 4 °C. The suspension was then applied to a column. The beads on the column were then washed with 8 ml of wash buffer (50 mM Tris, 100 mM NaCl, 20 mM imidazole, pH 8). The protein was eluted by 4 ml elution buffer I (50 mM Tris, 100 mM NaCl, 100 mM imidazole, pH 8) and 4 ml of elution buffer II (50 mM Tris, 100 mM NaCl, 150 mM imidazole, pH 8). All elutions were then rebuffered to 50 mM Tris pH 8, 100 mM NaCl, using Amicon Ultra-4 10k centrifugal filter tubes. The protein binding was measured by the microscale thermophoresis technique on a Monolith NT.115 instrument (Nanotemper). Protein (10 nM) in 50 mM HEPES buffer with 50 mM NaCl was labelled with Red-NHS dye NT-647. Maximum concentration of titrated protein was 50 μM.

**Black lipid membrane measurements**. Planar lipid membrane experiments were performed as described previously[92]. The electrolyte solution contained 1 M KCl and 10 mM Tris-HCl (pH 7.4). In all, 5 μl of purified protein (80 ng/ml) was mixed with 1500 μl KCl and was added to the *cis* compartment with a positive electrode, whereas the *trans* compartment was grounded. Planar lipid membrane was formed across a 0.5-mm aperture by painting the *E. coli* polar lipids extract (3% wt/vol, Avanti Polar Lipids, Alabaster, USA) dissolved in *n*-decan and butanol (9:1). The membrane current was registered using Ag/AgCl electrodes with salt bridges connected to an LCA-200-10G amplifier (Femto, Germany) and digitised with a KPCI-3108 16-Bit A/D card (Keithley Instruments, USA) with a 1-kHz sampling rate. Single-pore recordings were processed in the programme QuB[93]. The histogram of single-pore conductance ($n = 100$) was constructed by kernel density estimation (with the Gaussian kernel of 100-pS width) to overcome bin edge effects.

**In vitro protein translation and mitochondrial protein import**. The GoGspD and NgGspD genes were amplified from commercially synthesised templates (for primers used for PCR amplification of the coding sequences, see Supplementary Data 6) and cloned into pDHFR vector provided in the PURExpress In Vitro Protein Synthesis Kit (NEB). The translation into liposomes was done as described previously[94] and the output was analyzed by Blue Native PAGE using 2% digitonin and NativePAGE Novex 4–16% Bis-Tris Protein Gel (Thermo Fisher Scientific). For the in vitro mitochondrial protein import, the mitochondria were isolated from *S. cerevisiae* YPH499 according to the method described in Daum et al.[95]. The in vitro-translated NgGspD and Su9-DHFR chimeric construct[41] were incubated with mitochondria as described in Dolezal et al.[96]. The import reactions were incubated with 50 μg/ml of trypsin for 30 min on ice to remove unimported protein precursor.

**Testing protein interactions using two-hybrid systems**. Bacterial two-hybrid system (BACTH) analysis was performed as described before[97]. Gsp genes were amplified with specific primers (listed in Supplementary Data 6) and cloned into pKT25 and pUT18c plasmids. *E. coli* strain DHT1 competent cells were co-transformed with two plasmids with different combinations of Gsp genes. Co-transformants were selected on LB plates with ampicillin (Ap) (100 μg/mL) and kanamycin (Km; 25 μg/mL). Colonies were grown at 30 °C for 48–96 h. From each plate three colonies were picked, transferred to 1 mL of LB medium with Ap and Km, and grown overnight at 30 °C with shaking. Next day, precultures (0.25 mL) were inoculated to 5 mL of LB medium with Ap, Km and 1 mM IPTG. Cultures were grown with shaking at 30 °C to OD₆₀₀ of about 1–1.5. Bacteria (0.5 mL) were mixed with 0.5 mL of Z buffer and β-galactosidase activity was measured[98].

The yeast two-hybrid system (Y2H) was employed as described in Fields and Song[99]. Cells of *S. cerevisiae* strain AH109 were co-transformed with two plasmids (pGADT7, pGBKT7) with different combinations of Gsp genes. Co-transformants were selected on double-dropout SD-Leu/-Trp and triple-dropout SD-Leu/-Trp/-His plates. The colonies were grown for a few days. Positive colonies from the triple dropout were grown overnight at 30 °C with shaking and then the serial dilution test was performed on double- and triple-dropout plates.

**Reporting summary**. Further information on research design is available in the Nature Research Reporting Summary linked to this article.

## Data availability

All sequences of Gsp and Gcp proteins analysed in the study are provided in Supplementary Data 1. Gsp and Gcp genes extracted from an unpublished *Malawimonas*

*jakobiformis* genome assembly have been deposited at GenBank with accession numbers MT460910-MT460938 (etc.). Raw genome sequencing reads from "*Malawimonas californiana*" and *Neovahlkampfia damariscottae* are available from NCBI under the BioProject PRJNA549687. The genome assembly of *N. damariscottae* has been deposited at GenBank with the accession number JABLTG000000000. The transcriptome assembly of *Gefionella okellyi*, the genome assembly and predicted proteins of "*Malawimonas californiana*", and partial genome assemblies of *Reclinomonas americana* are available from https://megasun.bch.umontreal.ca/papers/T2SS-2020/. The mass spectrometry proteomics data have been deposited in the ProteomeXchange Consortium via the PRIDE[100] partner repository with the dataset identifier PXD007764. Other relevant data (e.g. multiple sequence alignments used for phylogenetic analyses) are available from the authors upon request.

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

## Acknowledgements

We would like to thank Michelle Leger and Alastair Simpson for granting us access, prior to publication, to their transcriptomic data from *A. godoyi* and *G. okellyi*, respectively, data that were instrumental in annotating the Gsp and Gcp genes in our genome assemblies. This work was supported by Czech Science Foundation grants 13-29423S to P.D. and 18-18699S to M.E., and the KONTAKT II grant LH15253 provided by Ministry of Education, Youth and Sports of CR (MEYS) to P.D.; This work was also supported by MEYS within the National Sustainability Program II (Project BIOCEV-FAR, LQ1604) the project BIOCEV (CZ.1.05/1.1.00/02.0109), and the project "Centre for research of pathogenicity and virulence of parasites" (No. CZ.02.1.01/0.0/0.0/16_019/0000759)

funded by European Regional Development Fund (ERDF) and MEYS and by Moore-Simons Project on the Origin of the Eukaryotic Cell to P.D. https://doi.org/10.37807/GBMF9738; The work in the OF laboratory was funded by the ANR-14-CE09-0004 grant. J.P. was supported by a grant from the Gordon and Betty Moore Foundation to ADT. This work was supported by The Ministry of Education, Youth and Sports from the Large Infrastructures for Research, Experimental Development and Innovations project "IT4Innovations National Supercomputing Center – LM2015070".

## Author contributions

L.H. planned and carried out the experiments, V.Ž. conceived the original idea and carried out the bioinformatics analyses, T.P. obtained the *N. damariscottae* genome sequence and carried out the genome and bioinformatic analyses, R.D. designed and carried out comparative genomic analyses, J.P. planned and carried out the experiments on *N. gruberi* mitochondrial proteome and analysed the data, A.M. carried out the experiments, Ve.K. carried out the experiments, M.V. carried out the experiments, L.M. carried out the experiments, L.V. carried out the experiments Vl.K. participated in genome sequencing and analysis, M.P. planned carried out the experiments, I.Č. participated in genome data acquisition, Kl.H. carried out the experiments, Z.V. carried out the experiments, Ka.H. analysed the proteome of *N. gruberi* mitochondria, M.W.G. contributed to the interpretation of the results and manuscript preparation, M.C. carried out the experiments and analyzed the data, I.G. designed and planned the experiments, O.F. designed, planned and carried out the experiments and analyzed the data, B.F.L. provided the genome data and analyses and contributed to manuscript preparation, Č.V. participated in genome data acquisition, A.D.T. designed and planned the experiments, M.E. conceived the idea, performed genomic analyses and wrote the manuscript, P.D. conceived the idea, designed and performed experiments and wrote the manuscript.

## Competing interests

The authors declare no competing interests.
