## [Peer Review File · Nature Communications]

Reviewers' comments:

Reviewer #1 (Remarks to the Author):

The findings presented by Horvathova and colleagues are interesting and important from an evolutionary perspective, showcasing the conservation of components of a bacterial Type 2 secretion system in mitochondria of distantly-related eukaryotes.

Major comment:

Evidence for mitochondrial localization of the Gsp subunits from *N. gruberi* and *G. okellyi* is presented. But the biological relevance of this conserved machinery is unclear, as function in mitochondrial protein secretion, and substrates of the machinery have not been uncovered in this study. One would expect this to be the case for Nature Communications. It is important to acknowledge the difficulty of working with the primitive eukaryotic organisms in question, however as it stands the proteins in question could simply be ancestral relics, as opposed to a functional secretion system. Thus, there is a general lack of biological data to support the major conclusion of a Type 2 secretion system being present in the mitochondria of these organisms.

Some minor comments that could be addressed to improve the manuscript:

[1] It is intriguing how a beta-barrel protein like GspD, which has a presequence, would be imported into the outer membrane. Could in vitro imports be performed to gauge some perspective on this?

[2] Are the N-terminal extensions described on page 5 functional mitochondrial targeting signals? For example, can they target GFP to mitochondria?

[3] Although mitochondrial localisation of the four eukaryotic Gsp proteins is shown in Figure 2, sub-mitochondrial location is not addressed. Indeed, Figure 7 is purely speculative and would need such biochemistry to be considered at all a possibility.

[4] Has GspD maintained bacterial or mitochondrial targeting elements for insertion into the outer membrane (i.e does it have a beta-signal)?

[5] In general the figures are very minimalistic for instance Figure 2, and it would be worthwhile considering moving some of the data from the supplement into the Figures.

Reviewer #2 (Remarks to the Author):

This paper reports an intriguing discovery of T2SS-like system in mitochondria of several eukaryotes. Initial sequence analysis identified 4 homologues of core components of the T2SS machinery. A subsequent phylogenetic analysis demonstrated presence of additional T2SS-related proteins, which represent either additional components of the system or potential substrates. Additionally, T2SS-related proteins were detected in the mitochondrial fraction. One of the T2SS core components, miGspD, or secretin, was found to be located in the mitochondria by immunofluorescent analysis. Partial characterization of miGspD confirmed it's ability to form homo oligomers in membranes.

Notes:

Figure 5C: some of the peptide sequences are either mis-aligned or mis-matched to the protein sequence.

Figure 5D: Labels (L,C,M) should be defined in the legend.

Figure 6A: Color of WD40 proteins appears purple, not red as stated in the legend.

Line 846: GenBank entries will need to be released.

Reviewer #3 (Remarks to the Author):

Bacteria rely on extracellular secretion for survival and virulence. Therefore, they have evolved complex protein secretion systems to transport specific substrates through their cell envelope. Although modern time mitochondria evolved from Gram-negative bacteria, it was commonly believed that the organelle did not preserved such dedicated secretion machineries.

In the present contribution, the authors report the bioinformatics detection of homologues of type 2 secretion system (T2SS) in some protist eukaryotes. They further show that these proteins are located in mitochondria and propose the formation of T2SS in mitochondria of these organisms. The authors suggest several additional proteins that might be linked to the core of the newly discovered T2SS.

The idea of a functional T2SS in mitochondria is novel and of great interest. However, despite the authors' claims, they do not provide any convincing support for the existence of a functional mitochondrial secretion machinery. Similarly, any insight regarding potential substrates of such a system is missing.

Major points:

1. The actual sequence homology of some of the putative mitochondrial Gsp homologues to representative bacterial Gsp proteins should be shown.
2. The mitochondria localization of the putative Gsp proteins should be studied in more detailed. The results shown in Fig. 2 and Suppl. Fig. 5 show that the proteins are associated with mitochondria but fail to demonstrate import into the organelle. In addition, in Fig. 2A it seems that the fusion proteins are located in additional sub-cellular structures beside mitochondria. (i) The authors should observed many cells (for example: three experiments of 100 cells each) and report in which percentage of the cells the proteins were located only to mitochondria. (ii) Western analysis of sub-mitochondrial and sub-cellular fractionation and demonstration of processing of the putative MTS should be shown. This can be done with the antibodies used in Fig. 2A. It will substantiate the authors' claims if they can show that GspD is embedded in the mitochondrial outer membrane.
3. The experiments shown in Fig. 4 suggest that GspD can form oligomers. However, the authors should provide evidence for pore formation either by structural analysis (for example: EM) or conductivity measurements.
4. The suggested interactions between mitochondrial GspF and GspG are based only on B2H. To substantiate this claim the authors should demonstrate such interactions by biochemical assays with recombinant purified proteins or proteins expressed in yeast cells.

Minor points:

- a. Lines 205-227: The explanation of which fraction was defined as mitochondria and which as peroxisome is not clear. Similarly confusing is the total number of 4198 proteins. In which fraction was this number of proteins found?
- b. It will be informative to show characterization by Western blotting of the antibodies used in Fig. 2A. How specific are they?
- c. Fig. 5D: the legend should explain what are L, C, and M.

Reviewer #4 (Remarks to the Author):

My name is Jeremy Wideman I am a proponent of open review. I have expertise in mitochondrial protein import, mitochondrial evolution, and eukaryote evolution and diversity. I declare some professional conflict in that I am in collaboration with Both the senior authors of this paper (Drs Dolezal and Elias), but on two unrelated projects. I am also currently at Dalhousie University in Halifax in the same department as Dr. Gray. I declared all of these conflicts to the editor before accepting the review. I believe that, by being aware of my subjectivity, I can effectively review and criticize this work.

Horváthová et al. present the first data, to my knowledge, of a mitochondrial export apparatus, specifically, the bacterial type II secretion system that has been passed down from the ancestral endosymbiont and has been sparsely retained in several disparate eukaryote lineages. This is an amazing finding and sheds light on the evolutionary processes leading to extant reduced mitochondria and hints at how mitochondria may have functioned in early eukaryotes.

From a technical point of view I see no obvious problems with the methods, conclusions or writing of this paper. It is all excellent work.

If I were reviewing this for Nature I would suggest that all the obvious T2SS components be expressed in yeast along with the best candidate for secretion so that the complex might be studied heterologously. However, as a Nature comms paper, I believe the novelty and rigour of the paper is more than sufficient to warrant publication.

I am however concerned about the replicability of the study since the authors have not (and will not) make their in house genomic and transcriptomic data publicly available.

According to Nature journal policy:

"An inherent principle of publication is that others should be able to replicate and build upon the authors' published claims. A condition of publication in a Nature Research journal is that authors are required to make materials, data, code, and associated protocols promptly available to readers without undue qualifications. Any restrictions on the availability of materials or information must be disclosed to the editors at the time of submission. Any restrictions must also be disclosed in the submitted manuscript."

Since whole-genome comparative analyses were done in order to cluster and identify the orthologous groups of T2SS and T2SS-related components it seems that these assemblies from very very very important eukaryotic lineages (Malawimonads, heteroloboseans, and Jakobids) should really be made available to the research community. Especially in today's sequencing climate it is becoming more and more trivial to sequence eukaryotic genomes. However, I believe the research community has largely refrained from sequencing these genomes out of politeness and a desire for collaboration and progress in our field rather than unnecessary competition.

Dependent upon how Nature editors enforce this policy and how one interprets the ability "to replicate and build upon the authors' published claims" will be the basis for if this paper is published in Nature Comms.

If the editors feel that this paper adheres to this policy, then I believe the paper should be published as is. If the editor(s) agrees with me that the assembled genomes should be made available so that this work can be effectively replicated and built upon, then after the genomes are made available, the paper should be published as is.

It should be noted that as a biologist that employs comparative genomic methods I am somewhat biased in my desire for open communication and sharing of resources. This is a bias that I am

proud of, but I feel it should be acknowledged.

Reviewers' comments:

Reviewer #1 (Remarks to the Author):

The findings presented by Horvathova and colleagues are interesting and important from an evolutionary perspective, showcasing the conservation of components of a bacterial Type 2 secretion system in mitochondria of distantly-related eukaryotes.

Major comment:

Evidence for mitochondrial localization of the Gsp subunits from *N. gruberi* and *G. okellyi* is presented. But the biological relevance of this conserved machinery is unclear, as function in mitochondrial protein secretion, and substrates of the machinery have not been uncovered in this study. One would expect this to be the case for Nature Communications. It is important to acknowledge the difficulty of working with the primitive eukaryotic organisms in question, however as it stands the proteins in questions could simply be ancestral relics, as opposed to a functional secretion system. Thus, there is a general lack of biological data to support the major conclusion of a Type 2 secretion system being present in the mitochondria of these organisms.

Authors' response: To a limited degree, we agree with the reviewer. We do admit that the central tenet of our study, that some mitochondria possess a functional protein secretion system, is not directly demonstrated by our experiments, as this is technically extremely challenging with regard to the nature of organisms exhibiting the system. It is important to appreciate that characterization of T2SS-driven protein secretion is very difficult even in highly tractable bacterial systems. Even in bacteria, it is impossible to identify secreted substrate based on protein sequence or even structure, since the molecular nature of secretion signals remains elusive. Still, we believe that the experimental tests described in the original submission and further expanded in the revised version of the manuscript collectively do support our hypothesis of the existence of a mitochondrial secretion system. We provide different lines of evidence supporting the mitochondrial localization of the eukaryotic Gsp homologs, we document oligomerisation of the mitochondrial GspG consistent with its presumed ability to assemble into a pseudopilus, and we demonstrate the ability of the GspD homolog to assemble into an oligomeric complex and insert in membranes *in vitro* making a membrane pore. These features are typical of bacterial secretin pores of the T2SS and in agreement with our model. In addition, we have a simple methodological argument favouring our interpretation. Positing that the eukaryotic mitochondrially-localized Gsp homologues assemble into a complex topologically and functionally equivalent to the bacterial T2SS is the most parsimonious interpretation of the undeniable existence of these proteins in particular eukaryotic taxa. In other words, this is the null hypothesis for the cellular role of eukaryotic Gsp proteins and in the absence of evidence contradicting it, it should be preferred over less parsimonious interpretations requiring extra assumptions, such as recruitment of the eukaryotic Gsp homologs for secretion-unrelated functions (which is perhaps what the reviewer had in mind when speaking about the proteins as "ancestral relics").

Some minor comments that could be addressed to improve the manuscript:

[1] It is intriguing how a beta-barrel protein like GspD, which has a presequence, would be imported into the outer membrane. Could *in vitro* imports be performed to gauge some perspective on this?

Authors' response: We subjected the *Naegleria* GspD protein to an *in vitro* protein import assay using isolated yeast mitochondria. The classical synthetic mitochondrial Su9-DHFR construct was used as a reference, i.e., as a marker for standard import pathway into the mitochondrial matrix. In contrast to Su9-DHFR, the import of GspD could not be blocked by the dissipation of the membrane potential by the AVO mix, which indicates its integration into the outer mitochondrial membrane. This result is presented in the revised manuscript as Fig. 4B. We also tested *in vitro* import of GspD into purified *N. gruberi* mitochondria along with *N. gruberi* marker proteins (e.g. Tom40, AAC, Nfu1). Unfortunately, none of the substrates was imported in a detectable quantity in the setting analogous to the *in vitro* import into the yeast mitochondria. We feel that establishing a functional *in vitro* import assay for *N. gruberi* mitochondria is a challenge for future work.

[2] Are the N-terminal extensions described on page 5 functional mitochondrial targeting signals? For example, can they target GFP to mitochondria?

Authors' response: The likely function of the extensions as mitochondrial targeting signals was supported by experiments included already in the original manuscript and commented on (page 5) as follows: "Moreover, the atypical MTSs of *N. gruberi* Gsp proteins were efficiently recognized by the yeast mitochondrial import machinery (Supplementary Fig. 5)." Admittedly, these experiments used translational fusions of the reporter fluorescent protein with full-length proteins investigated, so the mitochondrial localization in the heterologous system of *S. cerevisiae* could have been theoretically achieved by a mechanism not directly dependent on the N-terminal extension itself. To address the properties of the extensions in a more direct way, we expressed their fusions with mNeonGreen *Trypanosoma brucei*. Mitochondrial localization was observed in case of the N-terminal extension (first 159 amino acid residues) of *N. gruberi* GspG1 (Fig. S5B), supporting the function of the extension as a MTS. Unfortunately, analogous experiments with N-terminal extensions of *N. gruberi* GspE and GspF did not result in detectable protein expression.

[3] Although mitochondrial localisation of the four eukaryotic Gsp proteins is shown in Figure 2, sub-mitochondrial location is not addressed. Indeed, Figure 7 is purely speculative and would need such biochemistry to be considered at all a possibility.

Authors' response: We recognize the purely speculative nature of Fig. 7 (now Fig. 9), but we believe it has its place in the paper as a working hypothesis for the future investigations. In the figure we try to integrate the series of robustly documented facts presented in the paper with inferences dictated by parsimony reasoning, to come up with a plausible rationalization of the undeniable existence of eukaryotic homologues of the bacterial T2SS

components and the series of novel proteins with the same phyletic pattern. Thus, while we do not provide direct evidence for the sub-mitochondrial location of most of the proteins concerned, the topology proposed for the putative mitochondrial T2SS reflects is the most parsimonious inference reflecting the undisputed evolutionary continuity of the inner and outer mitochondrial membranes with the inner and outer membranes of Gram-negative eubacteria. It is worth mentioning that other mitochondrial protein translocases of the bacterial origin, such as Sam50 and Oxa1, do retain the localization of their bacterial ancestors, i.e., BamA and YidC, respectively, in the outer (Sam50/BamA) and inner (Oxa1/YidC) membrane.

[4] Has GspD maintained bacterial or mitochondrial targeting elements for insertion into the outer membrane (i.e does it have a beta-signal)?

Authors' response: GspD does not carry a C-terminal beta-signal (this information has been added to the text) but neither do the bacterial GspDs including the *Klebsiella* PulD, which has been extensively studied. The mechanism of the insertion and assembly of the oligomeric beta-barrels like GspD is different from the barrels composed of a single polypeptide. It has been shown in at least two independent studies that the bacterial GspD does not require the BAM complex for the pore assembly; see Collin et al. 2007 (<https://pubmed.ncbi.nlm.nih.gov/17542925/>). and Dunstan et al. 2015 (<https://pubmed.ncbi.nlm.nih.gov/25976323/>). Instead, upon forming multimers the so-called pre-pores insert spontaneously into liposomes/membranes via the AHL sequence, which is conserved in mitochondrial GspD.

[5] In general the figures are very minimalistic for instance Figure 2, and it would be worthwhile considering moving some of the data from the supplement into the Figures.

Authors' response: The figures in the main text have been substantially reorganized and expanded to include not only new results obtained during revision, but also some of the most important data previously presented only in the supplement.

Reviewer #2 (Remarks to the Author):

This paper reports an intriguing discovery of T2SS-like system in mitochondria of several eukaryotes. Initial sequence analysis identified 4 homologues of core components of the T2SS machinery. A subsequent phylogenetic analysis demonstrated presence of additional T2SS-related proteins, which represent either additional components of the system or potential substrates. Additionally, T2SS-related proteins were detected in the mitochondrial fraction. One of the T2SS core components, miGspD, or secretin, was found to be located in the mitochondria by immunofluorescent analysis. Partial characterization of miGspD confirmed it's ability to form homo oligomers in membranes.

Notes:

Figure 5C: some of the peptide sequences are either mis-aligned or mis-matched to

the protein sequence.

Figure 5D: Labels (L,C,M) should be defined in the legend.

Figure 6A: Color of WD40 proteins appears purple, not red as stated in the legend.

Line 846: GenBank entries will need to be released.

Authors' response: We have addressed all of these points while reorganizing the figures for the revised submission.

Reviewer #3 (Remarks to the Author):

Bacteria rely on extracellular secretion for survival and virulence. Therefore, they have evolved complex protein secretion systems to transport specific substrates through their cell envelope. Although modern time mitochondria evolved from Gram-negative bacteria, it was commonly believed that the organelle did not preserve such dedicated secretion machineries.

In the present contribution, the authors report the bioinformatics detection of homologues of type 2 secretion system (T2SS) in some protist eukaryotes. They further show that these proteins are located in mitochondria and propose the formation of T2SS in mitochondria of these organisms. The authors suggest several additional proteins that might be linked to the core of the newly discovered T2SS.

The idea of a functional T2SS in mitochondria is novel and of great interest. However, despite the authors' claims, they do not provide any convincing support for the existence of a functional mitochondrial secretion machinery. Similarly, any insight regarding potential substrates of such a system is missing.

Authors' response: Our answer here is essentially the same as our reply to similar reservations expressed by the Reviewer #1, above. Briefly, we believe that positing the existence of a functional mitochondrial secretion machinery is the most appropriate working hypothesis, because it is the most parsimonious explanation of the very existence of the eukaryotic Gsp homologs. Crucially, experiments we have carried out are all in agreement with this null hypothesis. Furthermore, we respectfully disagree with the reviewer's statement that "any insight regarding potential substrates of such a system is missing". By this statement, he/she ignores a key part of our study that does provide specific insights concerning the potential substrate(s) of the putative mitochondrial T2SS. By this we refer to the phylogenetic profiling analyses that revealed the existence of a series of novel proteins exactly co-occurring in eukaryotes with the core T2SS components (Gcp proteins). Note that the identification of both the mitochondrial T2SS components and most Gcp proteins in another eukaryote lineage, Hemimastigophora, provides further strong evidence for the functional association of the mitochondrial T2SS and Gcp proteins. It is logical to assume that the potential substrate(s) of the mitochondrial T2SS co-occur with the putative secretion machinery, so a hypothesis that some of the Gcp proteins are the sought-after substrates makes perfect sense to us. In the absence of any reasonable candidate for the substrate encoded directly by the mitochondrial genome, one has to rationalize why the protein should first enter the mitochondrion to subsequently be exported from it by the T2SS machinery. A specific modification or processing step taking place in the mitochondrion is a perfectly logical explanation – in fact the only one we can suggest. In this light,

the Gcp6 and Gcp12 proteins with the clusters of absolutely conserved cysteine and histidine residues, which are typical for proteins known to bear covalently attached prosthetic groups, are ideal candidates for the sought-after substrates. It is important to note that substrates containing such cofactors need to be secreted in the folded state, and T2SS is precisely the system specialized in secretion of folded proteins. Hence, contrary to the reviewer's opinion we believe that we have indeed provided relevant insights regarding potential substrates of the mitochondrial T2SS.

Major points:

1. The actual sequence homology of some of the putative mitochondrial Gsp homologues to representative bacterial Gsp proteins should be shown.

Authors' response: Alignments of the two proteins investigated in the greatest detail, GspD and GspG, including bacterial homologs, have been integrated into the main figures (Fig. 4 and 6). Alignments of the other proteins are provided as supplementary figures.

2. The mitochondria localization of the putative Gsp proteins should be studied in more detailed. The results shown in Fig. 2 and Suppl. Fig. 5 show that the proteins are associated with mitochondria but fail to demonstrate import into the organelle. In addition, in Fig. 2A it seems that the fusion proteins are located in additional sub-cellular structures beside mitochondria.

(i) The authors should observed many cells (for example: three experiments of 100 cells each) and report in which percentage of the cells the proteins were located only to mitochondria.

Authors' response: We strove to obtain more robust experimental evidence for the mitochondrial localization of the proteins studied. This meant that we had to raise a new set of specific polyclonal antibodies, as those used in the original submission ceased to work during relocation of the laboratory. Hence, we invested a great deal of effort into purification of new antigens, i.e., *N. gruberi* GspD, GspE, GspF, GspG1, GspG2 and GspEL2, for antibody production. Due to low immunogenicity of the prepared antigens, this difficult work — one of the main reasons for the delayed re-submission of the manuscript — eventually yielded useful polyclonal antibodies for only two of the proteins, GspG1 and GspEL2 (new Supplementary Fig. 6 demonstrates the specificity of the antibodies). Since we could not replicate the previous immunofluorescence experiments to ensure reproducibility, in the revised manuscript we report only the results of experiments with the newly raised anti-GspG1 and anti-GspEL2 antibodies, which are shown in Fig. 3D and which support mitochondrial localization of both GspG1 and GspEL2. Here, we do not show the statistics of the protein localization of the larger set of cells, as these new antibodies clearly labelled *N. gruberi* mitochondria. Fig. 3C presents a new experiment showing that the proteins recognized by the antibodies are present in an Optiprep-purified mitochondrial fraction, together with mitochondrial marker proteins. We note that the mitochondrial localization of the studied proteins is supported also by experiments with their heterologous expression

in the yeast, which were already included in the initial submission. To further test the conjecture of mitochondrial localization, we additionally tried to express the core T2SS components of *N. gruberi* and *G. okellyi* in *Trypanosoma brucei*. Of these we could detect only the expression of GoGspD, GoGspG2 and NgGspG1 (which are also the two most abundant T2SS components in bacterial systems), with the latter two proteins exhibiting the expected localization (Fig. 3C, Supplementary Fig. 5A).

(ii) Western analysis of sub-mitochondrial and sub-cellular fractionation and demonstration of processing of the putative MTS should be shown. This can be done with the antibodies used in Fig. 2A. It will substantiate the authors' claims if they can show that GspD is embedded in the mitochondrial outer membrane.

Authors' response: While we were able to detect the presence of GspG1 and GspEL2 in the Optiprep-purified mitochondria, our attempts at mitochondrial sub-fractionation were not successful. Neither the protocols based on digitonin treatment nor the incubation of mitochondria in hypotonic conditions provided reproducible results. In order to provide support for the presence of GspD in the outer mitochondrial membrane, we employed an alternative approach whereby we tested import of NgGspD *in vitro* into isolated yeast mitochondria (see also the response to point [1] of the reviewer #1). By using the classical mitochondrial matrix reporter (Su9-DHFR) as a control, the assay indicated membrane potential-independent accumulation of GspD in the mitochondria — a behaviour typical for outer membrane proteins. These new results have been incorporated into the revised manuscript.

3. The experiments shown in Fig. 4 suggest that GspD can form oligomers. However, the authors should provide evidence for pore formation either by structural analysis (for example: EM) or conductivity measurements.

Authors' response: To this aim, we purified GoGspD and tested its ability to form membrane pores by both electrophysiology and EM. The conductivity measurements showed the formation of highly stable open pores (data included as a part of Fig. 5), typical of bacterial secretins (Disconzi et al. 2014; <https://pubmed.ncbi.nlm.nih.gov/24142256/>). Furthermore, as now described in the revised manuscript, we detected the presence of pores of non-uniform sizes, indicating a possible capability of mitochondrial GspD to oligomerize into complexes of different stoichiometry.

4. The suggested interactions between mitochondrial GspF and GspG are based only on B2H. To substantiate this claim the authors should demonstrate such interactions by biochemical assays with recombinant purified proteins or proteins expressed in yeast cells.

Authors' response: We have now been able to demonstrate specific protein-protein interactions between GspG1 proteins in an *in vitro* assay; the data are now part of Fig. 7. Purification of GspF did not result in a sufficient amount of

the full-length protein, so unfortunately the *in vitro* experiment could not be carried out for this protein. GspF is a polytopic membrane protein of low abundance and very difficult to purify even in bacterial T2SSs.

Minor points:

a. Lines 205-227: The explanation of which fraction was defined as mitochondria and which as peroxisome is not clear. Similarly confusing is the total number of 4198 proteins. In which fraction was this number of proteins found?

Authors' response: We have added the information about the sub-fraction most enriched for mitochondria. However, the proteins were not classified as mitochondrial or peroxisomal based on their simple occurrence in different fractions, but based on their relative abundance in the three sub-fractions analyzed, as explained in the manuscript. The total number of 4198 proteins reflects proteins identified in all three sub-fractions combined, which has now been clarified in the text.

b. It will be informative to show characterization by Western blotting of the antibodies used in Fig. 2A. How specific are they?

Authors' response: In the revised manuscript we present results obtained only with newly raised antibodies. Their specificity is demonstrated by blots shown in Supplementary Fig. 6.

c. Fig. 5D: the legend should explain what are L, C, and M.

Authors' response: This figure has been omitted from the revised manuscript.

Reviewer #4 (Remarks to the Author):

My name is Jeremy Wideman I am a proponent of open review. I have expertise in mitochondrial protein import, mitochondrial evolution, and eukaryote evolution and diversity. I declare some professional conflict in that I am in collaboration with Both the senior authors of this paper (Drs Dolezal and Elias), but on two unrelated projects. I am also currently at Dalhousie University in Halifax in the same department as Dr. Gray. I declared all of these conflicts to the editor before accepting the review. I believe that, by being aware of my subjectivity, I can effectively review and criticize this work.

Horváthová et al. present the first data, to my knowledge, of a mitochondrial export apparatus, specifically, the bacterial type II secretion system that has been passed down from the ancestral endosymbiont and has been sparsely retained in several disparate eukaryote lineages. This is an amazing finding and sheds light on the evolutionary processes leading to extant reduced mitochondria and hints at how mitochondria may have functioned in early eukaryotes.

From a technical point of view I see no obvious problems with the methods, conclusions or writing of this paper. It is all excellent work.

Authors' response: Thank you for the kind words about our work.

If I were reviewing this for Nature I would suggest that all the obvious T2SS components be expressed in yeast along with the best candidate for secretion so that the complex might be studied heterologously. However, as a Nature comms paper, I believe the novelty and rigour of the paper is more than sufficient to warrant publication.

Authors' response: We would argue that attempts to reconstitute the whole system including the substrate in a heterologous system would be extremely challenging and most likely technically impossible, given the complexity of the system and lack of detailed knowledge of the function and significance of its individual components. In addition, we cannot rule out the possibility that the functionality of the system depends on factors that are not restricted to the species bearing the system (hence are not among the Gsp and Gcp proteins).

I am however concerned about the replicability of the study since the authors have not (and will not) make their in house genomic and transcriptomic data publicly available.

According to Nature journal policy:

"An inherent principle of publication is that others should be able to replicate and build upon the authors' published claims. A condition of publication in a Nature Research journal is that authors are required to make materials, data, code, and associated protocols promptly available to readers without undue qualifications. Any restrictions on the availability of materials or information must be disclosed to the editors at the time of submission. Any restrictions must also be disclosed in the submitted manuscript."

Since whole-genome comparative analyses were done in order to cluster and identify the orthologous groups of T2SS and T2SS-related components it seems that these assemblies from very very very important eukaryotic lineages (Malawimonads, heteroloboseans, and Jakobids) should really be made available to the research community. Especially in today's sequencing climate it is becoming more and more trivial to sequence eukaryotic genomes. However, I believe the research community has largely refrained from sequencing these genomes out of politeness and a desire for collaboration and progress in our field rather than unnecessary competition.

Dependent upon how Nature editors enforce this policy and how one interprets the ability "to replicate and build upon the authors' published claims" will be the basis for if this paper is published in Nature Comms.

If the editors feel that this paper adheres to this policy, then I believe the paper should be published as is. If the editor(s) agrees with me that the assembled genomes should be made available so that this work can be effectively replicated and built upon, then after the genomes are made available, the paper should be published as is.

It should be noted that as a biologist that employs comparative genomic methods I am somewhat biased in my desire for open communication and sharing of resources. This is a bias that I am proud of, but I feel it should be acknowledged.

Authors' response: We do appreciate the reviewer's criticism and one of the reasons for the delayed delivery of the revised manuscript has been to properly address this criticism. The current situation concerning the accessibility of the data used in the study can be summarized as follows:

1. Since our initial submission, the genome sequence and annotation of the jakobid *Andalucia godoyi* has been fully released as part of another paper co-authored by some of us (Gray et al., BMC Biology 2020).
2. The genomic data from the jakobid *Reclinomonas americana* used in this study, i.e. three different very incomplete genome assembly versions, have been deposited at a publicly available website indicated in the manuscript.
3. The genome sequence of the heterolobosean *Neovahlkampfia damariscottae* has been released to GenBank (with the accession number JABLTG000000000) and is now claimed as an additional significant result of the present study (the description of the sequencing and assembly has been integrated into Supplementary Methods).
4. The draft genome sequence and annotation of the malawimonad *Malawimonas californiana* has been deposited at a publicly available website indicated in the manuscript. Providing the data to the community is also considered by us to be an important extra outcome of the present study (although we explicitly comment on the limitations of the data, i.e., the presence of bacterial contamination and sequencing errors in homopolymeric regions; see Supplementary Methods). Releasing this genome is important to let anybody check the surprising absence of Gsp and Gcp genes in this malawimonad (contrasting with the presence of full sets of these genes in its two relatives).
5. We provide on a publicly accessible website a transcriptome assembly for the malawimonad *Gefionella okellyi* obtained from publicly available RNAseq reads generated by others. This assembly includes transcripts corresponding to all Gsp and Gcp genes of this species.
6. We have deposited to GenBank Gsp and Gcp genes individually extracted from our unpublished genome assembly for the malawimonad *Malawimonas jakobiformis*.

Altogether, we have now made available sequences most of the sequence data we have employed in our comparative analyses. We prefer not to release the full genome assemblies and annotations of *G. okellyi* and *M. jakobiformis*, as generating the data has been an important achievement in its own right, one that we believe deserves to be acknowledged by a separate publication of the data. We also do not release the transcriptome assemblies for some heteroloboseans that do not have the Gsp and Gcp genes, as they are not critical for our conclusions and are part of a separate broadly conceived study dedicated to various questions of heterolobosean biology and evolution. Generating genome sequence data from difficult-to-grow organisms is still a challenge, especially for small labs like ours, so we hope our preference to

reserve some of the assemblies for separate studies will be understood by the reviewer as well as the journal.

REVIEWER COMMENTS

Reviewer #2 (Remarks to the Author):

The revised manuscript by Horvathova et al. demonstrates presence of homologs of the bacterial type 2 secretions system components in mitochondria of a number of eukaryotic species. In addition to bioinformatic analysis, authors provide new experimental data supporting functional similarities between bacterial T2SS and miT2SS. The experiments showing oligomerization of GspG and GspD homologs, as well as pore-forming properties of GspD homolog indicate that miT2SS might function similar to bacterial T2SSs. Most of the points were addressed during revision.

Some minor points:

Page 3, line 4. Reference #19 did not present evidence of GspD oligomerization.

Page 3. Reference #18 is not very suitable for hexamerization of GspE and assembly of cytoplasmic platform.

Reviewer #3 (Remarks to the Author):

This is a revised version of a manuscript that I reviewed two years ago to Nature Comm. Despite the long period that passed since the original submission, I do not feel that the authors addressed properly my main comments on their original submission.

1. The authors still do not provide any evidence for a functional T2SS-like system in mitochondria and along this line, do not deliver any indication for substrates of such a putative system.

2. Furthermore, the evidences supporting mitochondrial and intra-mitochondrial location of the proteins is often not convincing or lacks critical controls. In details:

(i) Fluorescence microscopy is shown for only part of the proteins and even in these cases only ONE cell for each protein is shown (Fig. 3A, B, and D) without providing any statistic or at least a wider field where more cells can be observed.

(ii) The sub-cellular fractionation (Suppl. Fig. 6 and Fig. 3C) was done without proper controls - What do fractions #2 and 3 represent? In which fraction one should expect other membrane containing compartments (like peroxisomes or ER)? At its current state, fraction 3 can simply represent a mixture of many compartments.

(iii) According to mass spectrometry, some Gsp proteins are in a cluster with potential peroxisome proteins (Fig. 3E).

(iv) The assay for the in import of NgGspD into the mitochondrial outer membrane is not analyzed properly (Fig. 4B). Similar results would be obtained upon simple unspecific adherence of the hydrophobic protein to the surface of the organelle. Specific assays like carbonate extraction, resistance to externally added proteases, and/or formation of unique proteolytic fragment should be utilized.

3. Also the proposed formation of a well-defined pore structure is not convincing. The results of Fig. 4D show that GoGspD can form in vitro oligomers in the presence of liposomes. However, the data does not provide evidence for actual insertion into the membrane. This critic is supported by the observation that similar oligomeric structures are observed also with the recombinant protein in the presence of detergent, which cannot form bilayer membrane structures (Fig. 5B). Moreover, the high variability in the size of the pore (Fig. 5B and C) raises questions about its physiological relevance.

Minor points:

a. Mw markers are missing in many of the presented gels (for example in Fig. 7A).

b. Fig. 8C: The authors show mitochondrial localization of one protein namely, GoGpc12. Does it mean that all other 15 Gpc proteins could not be localized to mitochondria?

Reviewer #4 (Remarks to the Author):

I was very happy with the work that was done in the original review. My issues with data availability have been dealt with in large part. I think the work in the revision further merits publication in Nature Comms.

I worry that some of the requests from other reviewers are too big of an ask, so I will voice my opinions here. The organisms that are being investigated are not model systems and biochemical and cell biological experiments of in vivo functions are nearly impossible at the moment. Proving that the T2SS functions as a T2SS in eukaryotes is worthy of a career, not a single paper. Biochemists and cell biologists often forget that not every organism is a model organism. This paper from the Dolezal lab is a great step forward for investigation into mitochondrial secretion, and deserves to be published in Nature Comms.

REVIEWER COMMENTS

Reviewer #2 (Remarks to the Author):

The revised manuscript by Horvathova et al. demonstrates presence of homologs of the bacterial type 2 secretions system components in mitochondria of a number of eukaryotic species. In addition to bioinformatic analysis, authors provide new experimental data supporting functional similarities between bacterial T2SS and miT2SS. The experiments showing oligomerization of GspG and GspD homologs, as well as pore-forming properties of GspD homolog indicate that miT2SS might function similar to bacterial T2SSs. Most of the points were addressed during revision.

We thank the reviewer for the comments.

Some minor points:

Page 3, line 4. Reference #19 did not present evidence of GspD oligomerization.

Page 3. Reference #18 is not very suitable for hexamerization of GspE and assembly of cytoplasmic platform.

These are relevant points. We have replaced the references by more appropriate ones, ensuring that they provide real support to the statements in the respective sentences.

Reviewer #3 (Remarks to the Author):

This is a revised version of a manuscript that I reviewed two years ago to Nature Comm. Despite the long period that passed since the original submission, I do not feel that the authors addressed properly my main comments on their original submission.

1. The authors still do not provide any evidence for a functional T2SS-like system in mitochondria and along this line, do not deliver any indication for substrates of such a putative system.

We thank the reviewer for his/her comments and we agree with some of the points raised. Concerning the first critical point of the reviewer, we agree we have not directly demonstrated that the homologs of the bacterial T2SS components assemble into a functional protein translocation system in mitochondria possessing them. The key thing is we do not claim otherwise in manuscript. Hence, the problem boils down to the question is our work is publishable despite the lack of direct evidence for the functionality of the system. We strongly believe so, and we maintain that our manuscript delivers an exciting and rich story that will attract a lot of attention of a broad community or researchers and will stimulate multiple further studies: everybody will be invited to attempt overcoming the principal technical obstacles that have prevented us to gain a more complete picture of the system we are strongly persuaded exists in mitochondria of some eukaryotes.

2. Furthermore, the evidences supporting mitochondrial and intra-mitochondrial location of the proteins is often not convincing or lacks critical controls. In details: (i) Fluorescence microscopy is shown for only part of the proteins and even in these cases only ONE cell for each protein is shown (Fig. 3A, B, and D) without providing any statistic or at least a wider field where more cells can be observed.

We agree that additional images were important specifically for the immunofluorescent detection of *N. gruberi* GspG1 and GspEN2A proteins. We have included new images as part of the new Supplementary Figure 7.

(ii) The sub-cellular fractionation (Suppl. Fig. 6 and Fig. 3C) was done without proper controls - What do fractions #2 and 3 represent? In which fraction one should expect other membrane containing compartments (like peroxisomes or ER)? At its current state, fraction 3 can simply represent a mixture of many compartments.

We agree that the analysis of sub-cellular fractionation is not exhaustive, but we believe that it is the combination of the employed methods that strongly argues against the possibility of having a mixture of all compartments in the fraction 3. First, during the fractionation, the high speed pellet (HSP) fraction was obtained, which contained all sedimentable membrane-bounded compartments. The HSP was further separated into three separate fractions using density gradient (as observed on the Supplementary figure 8). Hence, all three bands correspond to a membrane-bounded compartments with different densities. After the comparable protein loading (shown in Fig. 3C), only the fraction 3 was positive for the mitochondrial proteins tested, showing that it is indeed a fraction most enriched for mitochondria. Moreover, in the label-free proteomic analysis, it was just the ratio of mitochondrial marker proteins (their spectra intensities) between fractions 1, 2 and 3 that defined the mitochondrial proteome and corroborated that the fraction 3 is most enriched for mitochondria. We believe that the procedure of assigning the different proteins to mitochondria or peroxisomes is clearly described in Methods. It is worth mentioning that due to the limited toolbox available for *N. gruberi*, there are no specific antibodies raised against the ER or the peroxisomal markers, and our experiments with commercial heterologous antibodies (anti-KDEL, anti-human calcineurin) did not show any specific labelling when applied on extracts from *N. gruberi*.

(iii) According to mass spectrometry, some Gsp proteins are in a cluster with potential peroxisome proteins (Fig. 3E).

Yes, we agree that some proteins of interest in Fig. 3E group with the peroxisomal proteins, but these are mainly Gcp proteins, which could actually function within peroxisome, as discussed in the manuscript. The core Gsp proteins are found within the mitochondrial cluster.

(iv) The assay for the import of NgGspD into the mitochondrial outer membrane is not analyzed properly (Fig. 4B). Similar results would be obtained upon simple unspecific adherence of the hydrophobic protein to the surface of the organelle.

Specific assays like carbonate extraction, resistance to externally added proteases, and/or formation of unique proteolytic fragment should be utilized.

All *in vitro* import reactions (GspD and Su9-DHFR) followed the protocol described in ref. #92, as stated in the manuscript. Thus, although not mentioned explicitly, these reactions included treatment with trypsin (50 µg/ml) after the import, removing proteins merely adhered to the surface of the membrane. Hence, the observed signals on Fig. 4B correspond to trypsin-shaved reactions and the alternative explanations suggested by the reviewer are not relevant.

3. Also the proposed formation of a well-defined pore structure is not convincing. The results of Fig. 4D show that GoGspD can form *in vitro* oligomers in the presence of liposomes. However, the data does not provide evidence for actual insertion into the membrane. This critic is supported by the observation that similar oligomeric structures are observed also with the recombinant protein in the presence of detergent, which cannot form bilayer membrane structures (Fig. 5B). Moreover, the high variability in the size of the pore (Fig. 5B and C) raises questions about its physiological relevance.

We think that our data convincingly show that mitochondrial GspD can form a pore. We did not have the ambition to show a “well-defined pore structure” (as mentioned by the reviewer). Our conclusion is supported by several independent lines of evidence: (i) Results of our homology modelling are compatible with notion that the mitochondrial GspD, like its bacterial homologs, can assemble into an oligomeric pore. (ii) Expression of mitochondrial GspD without a signal peptide results in quick bacterial death, compatible with it assembling a pore in the plasma membrane, as is expected for the variant lacking a signal for translocation to the periplasm. (iii) The recombinant protein assembles into similar oligomers on BN-PAGE and during gel filtration. In both cases, it is a detergent-solubilized complex (digitonin vs zwittergent); no bilayer can be expected on BN-PAGE either. (iv) Finally, the electrophysiology shows formation of very stable pores, which could hardly be possible due to a non-specific behavior.

Minor points:

a. Mw markers are missing in many of the presented gels (for example in Fig. 7A).

The missing markers have been added to revised Fig. 3C and 7A

b. Fig. 8C: The authors show mitochondrial localization of one protein namely, GoGpc12. Does it mean that all other 15 Gpc proteins could not be localized to mitochondria?

Of the various Gcp proteins, only Gcp12 was selected for specific investigations of its subcellular localization by expression in *T. brucei*. We have modified the wording of the sentence mentioning this experiment to make this easier to see from the text.

Reviewer #4 (Remarks to the Author):

I was very happy with the work that was done in the original review. My issues with data availability have been dealt with in large part. I think the work in the revision further merits publication in Nature Comms.

I worry that some of the requests from other reviewers are too big of an ask, so I will voice my opinions here. The organisms that are being investigated are not model systems and biochemical and cell biological experiments of in vivo functions are nearly impossible at the moment. Proving that the T2SS functions as a T2SS in eukaryotes is worthy of a career, not a single paper. Biochemists and cell biologists often forget that not every organism is a model organism. This paper from the Dolezal lab is a great step forward for investigation into mitochondrial secretion, and deserves to be published in Nature Comms.

We thank the reviewer for the encouraging comment.

REVIEWERS' COMMENTS

Reviewer #3 (Remarks to the Author):

The authors improved some points as compared to their previous version. However, I still find some problematic issues:

1. The authors agree that they could not demonstrate functionality or identify substrates of the suggested T2SS. Considering this, I find it misleading to use the name "mitochondrial T2SS system" when neither secretion nor substrates were demonstrated.

2. Fig. 3: It is still not clear what kind of proteins are in fractions #1 and #2 and whether fraction #3 represent only mitochondria. This can be found by mass spectrometry. How many of the non annotated proteins in Fraction #3 can be ER or POs proteins? At least it would be informative to know if T2SS-related proteins were found (in addition to Fraction #3), also in fractions #1 and #2. I could not find explanations in the text to this point.

3) Fig. 4D: The authors propose that the portion of NgGspD molecules that are resistant to trypsin treatment after the import reaction represent molecules that are embedded into the mitochondrial outer membrane. However, this portion might represent aggregated molecules. Hence, the authors should show that such a trypsin-resistant population is NOT detected when mitochondria are omitted from the import reaction. In addition, it would be good if the trypsin treatment would be mentioned in the text.

Response to reviewer's comments.

We thank the reviewer for these comments. Below we provide our specific answers including the description of the changes we have introduced into the manuscript to address the reviewer's points::

1. The authors agree that they could not demonstrate functionality or identify substrates of the suggested T2SS. Considering this, I find it misleading to use the name "mitochondrial T2SS system" when neither secretion nor substrates were demonstrated

Ad1

We are aware of the lack of decisive functional data that would demonstrate the "mitochondrial T2SS system" as the actual protein secretion machinery. Nevertheless, we argue that the term "mitochondrial T2SS", or miT2SS, is applicable even if the system we have identified does not transport proteins. Functional conservation is not the only criterion for naming things, evolutionary origin is equally valid. Nevertheless, to avoid any improper understanding, we double-checked the entire manuscript and have introduced a few minor linguistic changes (see the file with the manuscript version with all the changes tracked) to make it absolutely clear that the protein transport function of the mitochondrial T2SS, is presently only a hypothesis – one that is the most parsimonious and compatible with all the data that we present. Note also that we recognized this limitation already in the previous submission, where we changed the manuscript title from „Ancestral mitochondrial protein secretion machinery" to the current "Ancestral mitochondrial apparatus derived from the bacterial type II secretion system".

2. Fig. 3: It is still not clear what kind of proteins are in fractions #1 and #2 and whether fraction #3 represent only mitochondria. This can be found by mass spectrometry. How many of the non annotated proteins in Fraction #3 can be ER or POs proteins? At least it would be informative to know if T2SS-related proteins were found (in addition to Fraction #3), also in fractions #1 and #2. I could not find explanations in the text to this point.

Ad 2

In principle, all three fractions contained detectable amounts of mitochondrial as well as other cellular (ER-derived, peroxisomal etc.) proteins, but the different compartments exhibited a different level of enrichment in each fraction. The applied technique of label-free quantitative mass spectrometry generates large datasets of seemingly non-specific identifications on enriched (NOT purified) samples. It is the combination of these samples which makes the technique so powerful and specific. Firstly, the actual ratio of the detected intensities of well-known mitochondrial proteins among these three fractions was calculated. The proteins with a similar ratio of intensities were then classified according to principle component analysis as mitochondrial. Putative peroxisomal proteins were identified similarly by looking for proteins with a distribution profile across the sub-fractions similar to the distribution profile of well-established peroxisomal markers. We believe that the rationale of the analysis is described well in our manuscript, but we have double-checked the text and have made a slight modification in the section describing the results to avoid any

doubts about the principle of the method. We also realized that the labeling of Fig. 3C might be misleading, so we have relabeled the sub-fractions 1, 2, and 3 to Opt1015, Opt1520, and Opt2030, respectively, to ensure consistency with the Methods section.

Ad How many of the non annotated proteins in Fraction #3 can be ER or POs proteins?

The answer is that many of them. Of the total of 4,198 different proteins we identified across the three sub-fractions (as mentioned in the text), 946 were classified as mitochondrial and 78 as peroxisomal. The remaining 3,174 proteins represent hundreds of ER, cytosolic, nuclear and other proteins. At the moment we cannot easily compile a list of ER or other compartment-specific proteins, as this would require to re-analyze the entire dataset of 4,198 hits towards the compartment of interest. Crucially, we do not see the relevance of such an analysis, it would have no impact on the identification of mitochondrial or peroxisomal proteins or on the interpretation of the data in general.

Ad At least it would be informative to know if T2SS-related proteins were found (in addition to Fraction #3), also in fractions #1 and #2.

Yes, again, we identified many of them in these fractions, as mitochondria could be found in all three fractions (see above).

3) Fig. 4D: The authors propose that the portion of NgGspD molecules that are resistant to trypsin treatment after the import reaction represent molecules that are embedded into the mitochondrial outer membrane. However, this portion might represent aggregated molecules. Hence, the authors should show that such a trypsin-resistant population is NOT detected when mitochondria are omitted from the import reaction. In addition, it would be good if the trypsin treatment would be mentioned in the text.

We thank for this comment. Yes, we cannot completely exclude the possibility that the protease resistant protein species is a protein aggregate formed during the in vitro import reaction. However, the BN-PAGE in the same figure shows a digitonin-solubilized GspD complex when the protein is incubated with liposomes. The fact that protein aggregates do not form on the liposomal membrane argues against the aggregate formation on the mitochondrial membrane. We have added the information on the trypsin treatment to the Methods section, as this was indeed an unfortunate omission in the previous version of the manuscript.